# Inverse Reinforcement Learning via Inverse Optimization

## Abstract

The relationship between inverse reinforcement learning (IRL) and inverse optimization (IO) for Markov decision processes (MDPs) has been relatively underexplored in the literature, despite addressing the same problem. In this work, we revisit the relationship between the IO framework for MDPs, IRL, and apprenticeship learning (AL). We incorporate prior beliefs on the structure of the cost function into the IRL and AL problems, and demonstrate that the convex-analytic view of the AL formalism (Kamoutsi et al., 2021) emerges as a relaxation of our framework. Notably, the AL formalism is a special case in our framework when the regularization term is absent. Focusing on the suboptimal expert setting, we formulate the AL problem as a regularized min-max problem. The regularizer plays a key role in addressing the ill-posedness of IRL by guiding the search for plausible cost functions. To solve the resulting regularized-convex-concave-min-max problem, we use stochastic mirror descent (SMD) and establish convergence bounds for the proposed method. Numerical experiments highlight the critical role of regularization in learning cost vectors and apprentice policies.

## 1 Introduction

In scenarios where an agent must learn to navigate in a random or uncertain environment, it is a common practice to model the situation as a Markov decision process (MDP) and apply reinforcement learning (RL). The goal in RL is to find a policy that minimizes the total expected discounted cost for the agent. Usually, it is assumed that the cost function is known; however, specifying this function is difficult for most real-life scenarios (Ng & Russell, 2000). Moreover, an incorrect specification of the cost function can lead to unintended and potentially detrimental effects on the agent's behavior (Amodei et al., 2016; Hadfield-Menell et al., 2020). Consider the problem of driving: should the agent be rewarded for arriving quickly, safely, or cheaply, and how should the importance of each factor be balanced?

Inverse reinforcement learning (IRL) tackles this problem by reducing the work of manually designing the cost function and using observations of an expert agent's actions. Specifically, IRL aims to infer the cost function that the expert is optimizing based on recorded behavior and a model of the environment. Returning to the driving example, this approach involves observing an expert driver's behavior and deducing the underlying objective that guides their decisions. However, the goal extends beyond identifying the cost function; in many cases, there is a desire to emulate the expert's actions, much like a student assimilating knowledge from a mentor. For instance, when children learn to run, they are not explicitly given a cost function to optimize, but an expert shows them demonstrations of how they should run. Building on this idea, learning from demonstrations (LfD) and imitation learning (IL) seek to derive a policy that matches or surpasses the expert's performance.

IRL was first informally proposed by Russell (1998), and Ng & Russell (2000) introduced three algorithms for different scenarios: (1) when the policy, transition dynamics, and a finite state space are known; (2) when the state space is infinite; and (3) when the policy is unknown, but sample trajectories are available. Several methods have since been proposed, including a maximum margin approach (Ratliff et al., 2006), Bayesian frameworks (Ramachandran & Amir, 2007), and maximum entropy techniques (Ziebart et al., 2008). However, all of these methods rely on RL as a subroutine within an inner loop, leading to significant computational expenses.

Furthermore, the IRL problem is ill-posed (Ng & Russell, 2000), as multiple cost functions can explain an agent's behavior. This challenge has garnered increasing attention in the literature, with several works focusing on identifying the set of feasible cost functions that account for the expert's behavior (Metelli et al., 2021; Lindner et al., 2022). For our analysis, it is important to note that the work of Metelli et al. (2021; 2023); Lazzati et al. (2025) assumes access to a generative-model oracle for both the transition dynamics of the MDP and the expert's policy. A key finding in this body of work is that learning the feasible cost set is inefficient and infeasible for large state spaces, as the sample complexity scales heavily with the size of the state space.

In the context of LfD or IL, the literature often adopts the apprenticeship learning (AL) formalism proposed by Abbeel & Ng (2004), which assumes access to a set of expert demonstrations and that the unknown true cost function belongs to a specific class of functions. Consequently, this assumption requires identifying these basis functions in advance, which can be nontrivial. There are two main classes of cost functions considered: (1) linear combinations of known basis functions called features (Abbeel & Ng, 2004; Syed & Schapire, 2007; Ziebart et al., 2008) and (2) convex combinations of a set of vectors (Syed et al., 2008; Kamoutsi et al., 2021). Building on the AL formalism, Syed & Schapire (2007) presented a game-theoretic view of AL and solved it using a multiplicative weights algorithm. Later, Syed et al. (2008) proposed a linear programming approach to solve the AL problem without employing IRL or RL as a subroutine. This marked an initial step toward leveraging the tools of mathematical optimization to address the LfD problem. Following this direction, Kamoutsi et al. (2021) introduced a convex-analytic approach to the LfD problem within the AL formalism using a generative-model oracle for the MDP's transitions. They formulated a bilinear min-max problem using Lagrangian duality and solved it using stochastic mirror descent (SMD). Moreover, Ho & Ermon (2016) solved the LfD problem for a general class of cost functions $\mathcal{C} = \mathbb{R}^{\mathcal{S} \times \mathcal{A}}$, solving an entropy-regularized-min-max problem, and connected their approach with generative adversarial networks. Nevertheless, this min-max problem is nonconvex-nonconcave, limiting its theoretical understanding.

**Contribution.** We revisit IO's tools for IRL and present the inverse problem for estimating the cost function of an MDP given an optimal policy IRL-IO (Erkin et al., 2010; Chan et al., 2023) and incorporate prior beliefs on the structure of the cost function IRL-IO$_{\hat{c}}$. Through this approach, we revisit the proof that the inverse-feasible set of this inverse problem is equivalent to the dual problem derived by Kamoutsi et al. (2021) and extend it to a general class of cost functions. Furthermore, we propose a new problem for AL tailored for suboptimal experts IO-AL$_\alpha$ and provide results that allow us to interpret optimal solutions. Using Lagrangian duality, we derive a regularized-convex-concave-min-max problem RLfD$_\alpha$ for solving IO-AL$_\alpha$, which reduces to previous formulations (Kamoutsi et al., 2021) when the regularization term is null. Additionally, we show that the stochastic mirror descent algorithm proposed in Jin & Sidford (2020) to solve $\ell_\infty$-$\ell_1$ games naturally adapts to our problem, and we provide theoretical convergence bounds.

## 1.1   Notation

We denote the cardinality of a set $\mathcal{S}$ as $|\mathcal{S}|$. The probability simplex over $|\mathcal{S}|$ elements is given by $\Delta^{|\mathcal{S}|} = \left\{ \boldsymbol{x} \in \mathbb{R}^{|\mathcal{S}|} \mid x_i \geq 0, \sum_{i=1}^{|\mathcal{S}|} x_i = 1 \right\}$ and boxes are denoted by $\mathbb{B}_b^n = \{ \boldsymbol{x} \in \mathbb{R}^n \mid \|\boldsymbol{x}\|_\infty \leq b \}$. The canonical basis vectors are denoted by $\boldsymbol{e}^i = \{ \boldsymbol{x} \in \mathbb{R}^n \mid e_i = 1 \text{ and } e_j = 0 \,\forall j \neq i \}$. The Kronecker delta is denoted by $\delta_{ij}$. Component-wise multiplication between two vectors $\boldsymbol{x}, \boldsymbol{y}$ is denoted by $\boldsymbol{x} \circ \boldsymbol{y}$. The nonnegative real numbers are denoted by $\mathbb{R}_{\geq 0}$. Finally, we use the same notation as Kamoutsi et al. (2021), with minor modifications, to highlight the strong connections to their work.

## 2   Preliminaries and problem formulation

In this section, we establish the foundational concepts necessary for our study. We begin by defining the structure of infinite-horizon MDPs. We then introduce the IRL problem and discuss the LfD problem through the AL formalism. Finally, we provide an overview of IO and formally state our problem.

## 2.1 Infinite Horizon MDPs

A finite MDP is defined as a tuple $(\mathcal{S}, \mathcal{A}, P, \boldsymbol{\nu}_0, \boldsymbol{c}, \gamma)$ where $\mathcal{S}$ is a finite state space, $\mathcal{A}$ a finite action space, and $P$ is a transition law $P = (P(\cdot \mid s, a))_{s,a}$ where $P(\cdot \mid s, a) \in \Delta^{|\mathcal{S}|}$. The initial state distribution is denoted by $\boldsymbol{\nu}_0 \in \Delta^{|\mathcal{S}|}$ and satisfies $\boldsymbol{\nu}_0(s) > 0$ for every $s \in \mathcal{S}$. The cost vector is defined as $\boldsymbol{c} \in \mathcal{C} \subseteq \mathbb{B}_1^{|\mathcal{S}||\mathcal{A}|}$ and the discount factor is given by $\gamma \in (0, 1)$.

A *stationary Markov policy* is a collection of distributions, indexed by $s \in \mathcal{S}$ and denoted by $(\pi(\cdot \mid s))_{s \in \mathcal{S}}$, where $\pi(\cdot \mid s) \in \Delta^{|\mathcal{A}|}$. We denote the space of stationary Markov policies by $\Pi_0$. In this framework, the MDP begins with an initial state $s_0 \sim \boldsymbol{\nu}_0$. At each time-step $t$, where the current state is $s_t$: the agent selects an action according to $a_t \sim \pi(\cdot \mid s_t)$, the next state is determined by the transition law $s_{t+1} \sim P(\cdot \mid s_t, a_t)$, and a cost $c(s_t, a_t)$ is incurred. Note that in an infinite horizon model, the process continues indefinitely.

The *normalized value function* $\boldsymbol{V}_{\boldsymbol{c}}^{\pi} \in \mathbb{R}^{|\mathcal{S}|}$ of a policy $\pi$ given a cost $\boldsymbol{c}$ is given by

$$\boldsymbol{V}_{\boldsymbol{c}}^{\pi}(s) = (1 - \gamma) \mathbb{E}_s^{\pi} \left[ \sum_{t=0}^{\infty} \gamma^t c(s_t, a_t) \right]$$

where $\mathbb{E}_s^{\pi}[\cdot]$ denotes the expectation with respect to the trajectories generated by $\pi$ when starting from the state $s$. While we refer to it as a function, any function from a finite set to the reals can be naturally represented as a vector. The fundamental goal of RL is to find a policy $\pi$ such that the process $((s_t, a_t))_t$ minimizes the *total expected cost*:

$$\rho_{\boldsymbol{c}}^* = \min_{\pi \in \Pi_0} \rho_{\boldsymbol{c}}(\pi) \qquad (\text{RL}_{\boldsymbol{c}})$$

$$= \min_{\pi \in \Pi_0} (1 - \gamma) \mathbb{E}_{\boldsymbol{\nu}_0}^{\pi} \left[ \sum_{t=0}^{\infty} \gamma^t c(s_t, a_t) \right].$$

where $\rho_{\boldsymbol{c}}(\pi) = \langle \boldsymbol{\nu}_0, \boldsymbol{V}_{\boldsymbol{c}}^{\pi} \rangle$. Notice that we explicitly highlight the dependence of equation $\text{RL}_{\boldsymbol{c}}$ on the cost vector $\boldsymbol{c}$. Furthermore, we denote $\boldsymbol{V}_{\boldsymbol{c}}^*$ as the value function corresponding to the optimal policy for $\text{RL}_{\boldsymbol{c}}$.

The *normalized occupancy measure* $\boldsymbol{\mu}_{\pi} \in \Delta^{|\mathcal{S}||\mathcal{A}|}$ of a policy $\pi$ is defined as

$$\boldsymbol{\mu}_{\pi}(s, a) = (1 - \gamma) \sum_{t=0}^{\infty} \gamma^t \mathbb{P}_{\boldsymbol{\nu}_0}^{\pi}[s_t = s, \ a_t = a],$$

where $\mathbb{P}_{\boldsymbol{\nu}_0}^{\pi}[\cdot]$ represents the probability of an event when starting from $s \sim \boldsymbol{\nu}_0$ and following $\pi$. The occupancy measure of a state-action pair can be interpreted as the discounted visitation frequency of the pair when following a particular policy. Hence, we can also write $\rho_{\boldsymbol{c}}^* = \min_{\pi \in \Pi_0} \langle \boldsymbol{\mu}_{\pi}, \boldsymbol{c} \rangle$.

We define the transition matrix $\boldsymbol{P} \in \mathbb{R}^{|\mathcal{S}| \times |\mathcal{S}||\mathcal{A}|}$ where $\boldsymbol{P}_{s',(s,a)} = P(s' \mid s, a)$ and the polyhedron $\mathcal{F} = \{\boldsymbol{\mu} \in \mathbb{R}^{|\mathcal{S}||\mathcal{A}|} \mid \boldsymbol{T}_{\gamma} \boldsymbol{\mu} = \boldsymbol{\nu}_0, \ \boldsymbol{\mu} \geq \boldsymbol{0}\}$ where $\boldsymbol{T} \in \mathbb{R}^{|\mathcal{S}| \times |\mathcal{S}||\mathcal{A}|}$, $\boldsymbol{T}_{s',(s,a)} = \delta_{s',s} - \gamma \boldsymbol{P}_{s',(s,a)}$, and $\boldsymbol{T}_{\gamma} = \frac{1}{(1-\gamma)} \boldsymbol{T}$. An alternative expression for $\boldsymbol{T}_{\gamma}$ that is useful for computing gradient estimators is $\boldsymbol{T}_{\gamma} \boldsymbol{\mu} = \frac{1}{(1-\gamma)} (\boldsymbol{B} - \gamma \boldsymbol{P}) \boldsymbol{\mu}$ where $\boldsymbol{B}$ is a binary matrix that satisfies $\boldsymbol{B}_{s',(s,a)} = 1$ if $s' = s$ and $\boldsymbol{B}_{s',(s,a)} = 0$ otherwise.

**Proposition 1** (Puterman (1994)). *It holds that, $\mathcal{F} = \{\boldsymbol{\mu}_{\pi} \mid \pi \in \Pi_0\}$. For every $\pi \in \Pi_0$, we have that $\boldsymbol{\mu}_{\pi} \in \mathcal{F}$. Moreover, for every feasible solution $\boldsymbol{\mu} \in \mathcal{F}$, we can obtain a stationary Markov policy $\pi_{\boldsymbol{\mu}} \in \Pi_0$ by $\pi_{\boldsymbol{\mu}}(a \mid x) = \frac{\mu(x,a)}{\sum_{a' \in \mathcal{A}} \mu(x,a')}$. Then, the induced occupancy measure is exactly $\mu$.*

Proposition 1 provides a correspondence between the elements of $\mathcal{F}$ and occupancy measures given by stationary Markov policies. Note that the condition $\boldsymbol{T}_{\gamma} \boldsymbol{\mu} = \boldsymbol{\nu}_0$ can be interpreted as the Markov property of the process under $\mathbb{P}_{\boldsymbol{\nu}_0}^{\pi}[\cdot]$. Hence, the MDP linear programming approach consists of solving the MDP-$\text{P}_{\boldsymbol{c}}$ problem

$$\rho_{\boldsymbol{c}}^* = \min_{\boldsymbol{\mu} \in \Delta^{|\mathcal{S}||\mathcal{A}|}} \langle \boldsymbol{\mu}, \boldsymbol{c} \rangle$$

$$\text{s.t} \quad \boldsymbol{T}_{\gamma} \boldsymbol{\mu} = \boldsymbol{\nu}_0, \qquad (\text{MDP-P}_{\boldsymbol{c}})$$

$$\boldsymbol{\mu} \geq \boldsymbol{0}.$$

Note that the constraints enforce that $\boldsymbol{\mu} \in \mathcal{F}$, therefore an optimal $\boldsymbol{\mu}$ corresponds to an optimal policy. The corresponding dual problem is given by

$$\max_{\boldsymbol{u} \in \mathbb{R}^{|S|}} \{ \langle \boldsymbol{\nu}_0, \boldsymbol{u} \rangle \mid \boldsymbol{c} - \boldsymbol{T}_\gamma^\top \boldsymbol{u} \geq \boldsymbol{0} \}, \tag{MDP-D$_{\boldsymbol{c}}$}$$

where an optimal $\boldsymbol{u}$ represents the optimal value function $\boldsymbol{V}_{\boldsymbol{c}}^*$.

## 2.2 Inverse reinforcement learning

The IRL problem aims to uncover the true cost function that an expert agent is optimizing given some information about the expert's behavior: sample trajectories, its real policy, or an estimate of its policy (Ng & Russell, 2000). Formally, given an MDP without a cost vector and with access to information about an expert's policy $\pi_E$, which could be the actual policy, an estimate, or a set of demonstrations, the IRL problem is defined by the tuple $(\mathcal{S}, \mathcal{A}, P, \boldsymbol{\nu}_0, \pi_E, \gamma)$. The goal of the IRL problem is to determine a cost vector $\boldsymbol{c}$ for which the policy $\pi_E$ is optimal for RL$_{\boldsymbol{c}}$ within the MDP $(\mathcal{S}, \mathcal{A}, P, \boldsymbol{\nu}_0, \boldsymbol{c}, \gamma)$.

## 2.3 Learning from demonstrations and the apprenticeship learning formalism

The goal of learning from demonstrations is to learn a policy that matches or outperforms the expert's policy $\pi_E$ for an unknown true cost vector $\boldsymbol{c}_{\text{true}}$. The apprenticeship learning formalism (Abbeel & Ng, 2004) has been routinely used in literature for addressing the LfD problem. The AL formalism assumes that the unknown true cost function $\boldsymbol{c}_{\text{true}}$ belongs to a class of functions $\mathcal{C}$ and searches for a policy that solves the following min-max problem

$$\beta^* := \min_{\pi \in \Pi_0} \max_{\boldsymbol{c} \in \mathcal{C}} \langle \boldsymbol{\mu}_\pi, \boldsymbol{c} \rangle - \langle \boldsymbol{\mu}_{\pi_E}, \boldsymbol{c} \rangle = \min_{\pi \in \Pi_0} \max_{\boldsymbol{c} \in \mathcal{C}} \langle \boldsymbol{\mu}_\pi - \boldsymbol{\mu}_{\pi_E}, \boldsymbol{c} \rangle, \tag{LfD$_{\pi_E}$}$$

An optimal solution to LfD$_{\pi_E}$ is called an apprentice policy $\pi_A$ and satisfies

$$\langle \boldsymbol{\mu}_{\pi_A}, \boldsymbol{c}_{\text{true}} \rangle \leq \langle \boldsymbol{\mu}_{\pi_E}, \boldsymbol{c}_{\text{true}} \rangle + \beta^*.$$

In optimization-focused approaches to LfD, the $\boldsymbol{c}_{\text{true}}$ is assumed to belong to a convex hull

$$\mathcal{C} = \mathcal{C}_{\text{conv}} := \left\{ \boldsymbol{c}_{\boldsymbol{w}} := \sum_{i=1}^{n_c} w_i c_i \;\middle|\; w_i \geq 0, \sum_{i=1}^{n_c} w_i = 1 \right\}$$

(Syed et al., 2008; Kamoutsi et al., 2021) of a set of vectors $\{\boldsymbol{c}_i\}_{i=1}^{n_c} \subseteq \mathbb{R}^{|\mathcal{S}||\mathcal{A}|}$ where $\|\boldsymbol{c}_i\|_\infty \leq 1$ for each $i = 1, ..., n_c$. It is assumed that this set of vectors is known; however, in practice, an initial estimation step is required to determine this set, a task that is generally nontrivial.

## 2.4 A primer on inverse optimization

Inverse optimization is a mathematical framework that fits optimization models to decision data. Given an observed optimal solution, it seeks to learn the objectives and constraints of the underlying model. For example, IRL can be thought of as an inverse optimization problem, as it searches for the cost function that an optimal agent is optimizing.

Consider the general forward optimization problem FOP$_\theta$ for a given parameter $\theta$ in the parameter space $\Gamma$:

$$\min_{\boldsymbol{x} \in \mathbb{R}^n} \{ f(\boldsymbol{x}, \theta) \mid \boldsymbol{x} \in X(\theta) \}, \tag{FOP$_\theta$}$$

where $X(\theta)$ denotes the feasible set for $\boldsymbol{x}$, which depends on $\theta$. Given an optimal solution $\hat{\boldsymbol{x}}$, the inverse optimization problem consists of finding a $\theta^* \in \Gamma$ that makes $\hat{\boldsymbol{x}}$ optimal for FOP$_\theta$ with $\theta = \theta^*$ and is optimal in some way. For this purpose, define the optimal solution set $X^{\text{opt}}(\theta) := \arg\min_{\boldsymbol{x}} \{ f(\boldsymbol{x}, \theta) \mid \boldsymbol{x} \in X(\theta) \}$ and the inverse-feasible set $\Theta^{\text{inv}}(\hat{\boldsymbol{x}}) := \{ \theta \in \Gamma \mid \hat{\boldsymbol{x}} \in X^{\text{opt}}(\theta) \}$. Naturally, we want to find a $\theta \in \Theta^{\text{inv}}(\hat{\boldsymbol{x}})$, but rather than selecting an arbitrary $\theta$ from this set, we aim for one that minimizes a certain criterion.

Hence, the inverse optimization problem INV-OPT is defined as:

$$\min_{\theta \in \Gamma} \{F(\theta) \mid \theta \in \Theta^{\mathrm{inv}}(\hat{\boldsymbol{x}})\}, \tag{INV-OPT}$$

where $F$ should convey information about the quality of $\theta$ given some prior knowledge, and the search space $\Gamma$ should be appropriately chosen for each instance of the problem.

### 2.5 Our problem

Suppose that the environment is modeled as an MDP where only the state space $\mathcal{S}$, action space $\mathcal{A}$, and discount factor $\gamma$ are known. We assume that the learner has access to a generative-model oracle for the MDP's transition dynamics, as well as a generative-model oracle of an expert's occupancy measure $\boldsymbol{\mu}_{\pi_E}$ (not necessarily optimal), and a prior belief $\hat{\boldsymbol{c}}$ of the cost function the expert is trying to optimize for. We aim to learn a cost function $\boldsymbol{c}_A$ and an apprentice policy $\pi_A$, such that $\pi_A$ is optimal for $\mathrm{RL}_{\boldsymbol{c}_A}$, and $\boldsymbol{c}_A$ remains close to the prior $\hat{\boldsymbol{c}}$ while $\pi_A$ performs similarly to $\pi_E$ under $\boldsymbol{c}_A$ (see IO-AL$_\alpha$).

## 3 The inverse optimization viewpoint

In this section, we show how the IRL problem can be addressed with the tools of IO. We establish the equivalence between the AL problem formulation presented in Kamoutsi et al. (2021) and the inverse-feasible set of the problem IRL-IO. Additionally, we demonstrate how prior beliefs about the structure of the cost vector can be incorporated into both the IRL formulation IRL-IO$_{\hat{c}}$ and the AL setting IO-AL$_\alpha$.

### 3.1 IRL via IO

We will use the ideas of Subsection 2.4 applied to the forward optimization problem MDP-P$_{\boldsymbol{c}_{\mathrm{true}}}$, where the parameter $\theta$ corresponds to the true cost vector $\boldsymbol{c}_{\mathrm{true}}$ the expert is optimizing for. Note that we assume the existence of $\boldsymbol{c}_{\mathrm{true}}$ because the IRL problem assumes that the expert is optimal for some cost function. Therefore, let us suppose that $\boldsymbol{c}_{\mathrm{true}}$ lies in a convex class of cost functions $\mathcal{C}$ and that the expert's policy $\pi_E$ is optimal for $\mathrm{RL}_{\boldsymbol{c}_{\mathrm{true}}}$, which means that its corresponding occupancy measure $\boldsymbol{\mu}_{\pi_E}$ is optimal for MDP-P$_{\boldsymbol{c}_{\mathrm{true}}}$. The following proposition follows from complementary slackness for linear problems (Bertsimas & Tsitsiklis, 1997).

**Proposition 2** (Complementary slackness)**.** *An element $\boldsymbol{\mu}_\pi$ is an optimal solution to MDP-P$_{\boldsymbol{c}}$ if and only if there exists a vector $\boldsymbol{u} \in \mathbb{R}^{|\mathcal{S}|}$ such that $\boldsymbol{c} - \boldsymbol{T}_\gamma^\top \boldsymbol{u} \geq \boldsymbol{0}$ and $\langle \boldsymbol{\mu}_\pi, \boldsymbol{c} - \boldsymbol{T}_\gamma^\top \boldsymbol{u} \rangle = 0$.*

Remember that $\boldsymbol{u}$ is the dual variable for the equality constraint in MDP-P$_{\boldsymbol{c}}$ and represents the value function. Therefore, the inverse-feasible set for $\boldsymbol{\mu}_{\pi_E}$ consists of the cost functions in $\mathcal{C}$ for which such a $\boldsymbol{u}$ exists

$$\Theta^{\mathrm{inv}}(\boldsymbol{\mu}_{\pi_E}) := \{\boldsymbol{c} \in \mathcal{C} \mid \exists \boldsymbol{u} \in \mathbb{R}^{|S|} \, : \, \boldsymbol{c} - \boldsymbol{T}_\gamma^\top \boldsymbol{u} \geq \boldsymbol{0}, \, \langle \boldsymbol{\mu}_{\pi_E}, \boldsymbol{c} - \boldsymbol{T}_\gamma^\top \boldsymbol{u} \rangle = 0\}.$$

Substituting for the inverse-feasible set in INV-OPT and choosing an appropriate function $F$ for comparing cost vectors, we arrive to the inverse reinforcement learning problem through inverse optimization (Erkin et al., 2010; Chan et al., 2023)

$$\begin{aligned} \min_{\boldsymbol{c} \in \mathcal{C}, \boldsymbol{u} \in \mathbb{R}^{|S|}} \quad & F(\boldsymbol{c}) \\ \text{s.t} \quad & \boldsymbol{c} - \boldsymbol{T}_\gamma^\top \boldsymbol{u} \geq \boldsymbol{0}, \\ & \langle \boldsymbol{\mu}_{\pi_E}, \boldsymbol{c} - \boldsymbol{T}_\gamma^\top \boldsymbol{u} \rangle = 0. \end{aligned} \tag{IRL-IO}$$

### 3.2 Connections to LfD and the AL formalism

Kamoutsi et al. (2021) considered the LfD problem under the assumption that the true cost function $\boldsymbol{c}_{\mathrm{true}}$ belongs to the convex hull $\mathcal{C}_{\mathrm{conv}}$ of a given set of vectors. By applying an epigraphic transformation to

$\text{LfD}_{\pi_E}$, where the validity of this transformation depends on the previous assumption on $\boldsymbol{c}_{\text{true}}$, and deriving its dual, they arrived to the optimization problem $\text{D}_{\pi_E}$:

$$\max_{\boldsymbol{c},\boldsymbol{u}}\{\langle \boldsymbol{\mu}_{\pi_E}, \boldsymbol{T}_\gamma^\top \boldsymbol{u} - \boldsymbol{c}\rangle \mid \boldsymbol{c} \in \mathcal{C}_{\text{conv}}, \boldsymbol{c} - \boldsymbol{T}_\gamma^\top \boldsymbol{u} \geq \boldsymbol{0}\}. \qquad (\text{D}_{\pi_E})$$

They focus on this problem and optimize its unconstrained version derived through Lagrangian duality, where the dual variable corresponding to the constraint $\boldsymbol{c} - \boldsymbol{T}_\gamma^\top \boldsymbol{u} \geq \boldsymbol{0}$ represents the apprentice state-action visitation probability.

**Theorem 1** (cf. Proposition 2 in Kamoutsi et al. (2021)). *Suppose that $\boldsymbol{\mu}_{\pi_E}$ is an optimal solution for MDP-$P_{\boldsymbol{c}}$ where $\boldsymbol{c} \in \mathcal{C}$. Then the following equality holds:*

$$\Theta^{inv}(\boldsymbol{\mu}_{\pi_E}) = \Pi_1\left(\underset{(\boldsymbol{c},\boldsymbol{u})}{\arg\max}\{\langle \boldsymbol{\mu}_{\pi_E}, \boldsymbol{T}_\gamma^\top \boldsymbol{u} - \boldsymbol{c}\rangle \mid \boldsymbol{c} \in \mathcal{C}, \ \boldsymbol{c} - \boldsymbol{T}_\gamma^\top \boldsymbol{u} \geq 0\}\right)$$

*where $\Pi_1$ denotes the projection in the first component.*

This implies that the dual problem $\text{D}_{\pi_E}$ serves as an alternative representation of the inverse-feasible set $\Theta^{\text{inv}}(\boldsymbol{\mu}_{\pi_E})$. In contrast, in problem IRL-IO we choose an element within the inverse-feasible set that minimizes $F$. In this sense, under the assumption of expert's optimality and that $\boldsymbol{c}_{\text{true}} \in \mathcal{C}_{\text{conv}}$, the AL formalism finds an arbitrary element of the inverse-feasible set, whereas IRL-IO has a criterion for searching within this space and considers a general convex class of cost functions $\mathcal{C}$.

## 3.3 Incorporating prior beliefs

Suppose we are given a proxy cost vector $\hat{\boldsymbol{c}}$ that reflects our prior beliefs about the structure of the true cost vector, which are not necessarily accurate. Leveraging this information, we aim to guide the search within the inverse feasible set in problem IRL-IO. To this end, we project $\hat{\boldsymbol{c}}$ onto $\Theta^{\text{inv}}(\boldsymbol{\mu}_{\pi_E})$ by solving the following optimization problem:

$$\begin{aligned}
\min_{\boldsymbol{c}\in\mathcal{C},\boldsymbol{u}\in\mathbb{R}^{|S|}} \quad & \|\boldsymbol{c} - \hat{\boldsymbol{c}}\|_2^2 \\
\text{s.t} \quad & \boldsymbol{c} - \boldsymbol{T}_\gamma^\top \boldsymbol{u} \geq \boldsymbol{0}, \qquad\qquad (\text{IRL-IO}_{\hat{\boldsymbol{c}}}) \\
& \langle \boldsymbol{\mu}_{\pi_E}, \boldsymbol{c} - \boldsymbol{T}_\gamma^\top \boldsymbol{u}\rangle = 0.
\end{aligned}$$

Figure 1 illustrates the setting for problem IRL-IO$_{\hat{\boldsymbol{c}}}$. The polyhedral region in fuchsia represents the set of all occupancy measures $\mathcal{F}$. In particular, if we assume that the expert is optimal and the induced policy is deterministic, then $\boldsymbol{\mu}_{\pi_E}$ is a vertex of $\mathcal{F}$ and the true cost vector $\boldsymbol{c}_{\text{true}}$, in green, is within the inverse-feasible set $\Theta^{\text{inv}}(\boldsymbol{\mu}_{\pi_E})$. The proxy cost vector $\hat{\boldsymbol{c}}$, in red, is not necessarily inside the inverse-feasible set and will be projected onto the inverse-feasible set by solving Problem IRL-IO$_{\hat{\boldsymbol{c}}}$.

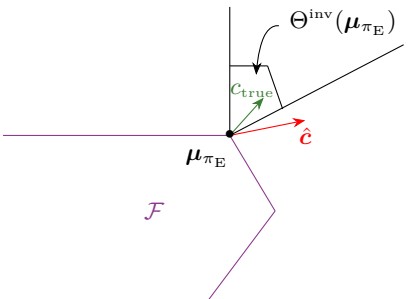

Figure 1: Illustration of the incorporation of $\hat{\boldsymbol{c}}$.

Since we are still selecting an element from the inverse-feasible set, we can derive results similar to those presented by Kamoutsi et al. (2021) for solutions to Problem IRL-IO$_{\hat{\boldsymbol{c}}}$ under the assumption of expert optimality. The following corollary follows directly from the proof of Theorem 1 (see Appendix A.2):

**Corollary 1** (Optimal expert). *Assume that $\pi_E$ is optimal for $RL_{c_{true}}$. A pair $(c_A, u_A)$ is optimal for Problem IRL-IO$_{\hat{c}}$ if and only if $\pi_E$ is optimal for $RL_{c_A}$ and $u_A = V_{c_A}^*$. In particular, the cost vector $c_A$ is the projection of $\hat{c}$ onto the inverse feasible set and $\pi_E$ is optimal for $c_A$.*

It is important to note that when the expert is suboptimal, Problem IRL-IO$_{\hat{c}}$ is infeasible, as the complementary slackness equality cannot be satisfied. To account for the possibility of suboptimal expert behavior, we can relax the complementary slackness condition in IRL-IO. Weighing the beliefs of the expert's optimality and the quality of the cost function estimate with parameter $\alpha \in \mathbb{R}_{\geq 0}$, we arrive to problem IO-AL$_\alpha$:

$$\min_{c \in \mathcal{C}, u \in \mathbb{R}^{|\mathcal{S}|}} \quad \alpha \|c - \hat{c}\|_2^2 + \langle \mu_{\pi_E}, c - T_\gamma^\top u \rangle$$
$$\text{s.t} \quad c - T_\gamma^\top u \geq 0. \tag{IO-AL$_\alpha$}$$

**Proposition 3** (Suboptimal expert). *A pair $(c_A, u_A)$ is optimal for IO-AL$_\alpha$ if and only if the apprentice policy $\pi_A$ is optimal for $RL_{c_A}$ and $u_A = V_{c_A}^*$. Furthermore, the optimal value corresponds to $\alpha \|c_A - \hat{c}\|_2^2 + \rho_{c_A}(\pi_E) - \rho_{c_A}(\pi_A)$.*

Proposition 3 states that the apprentice policy $\pi_A$, i.e. the dual variable for the constraint $c - T_\gamma^\top u \geq 0$, is optimal for the RL problem with cost vector $c_A$. Solutions to IO-AL$_\alpha$ can be viewed as a way to balance the distance between the cost vector $c_A$ and the estimate $\hat{c}$, while ensuring that the total expected cost of $\pi_E$ and $\pi_A$ are similar under $c_A$ and that $\pi_A$ is optimal for $RL_{c_A}$.

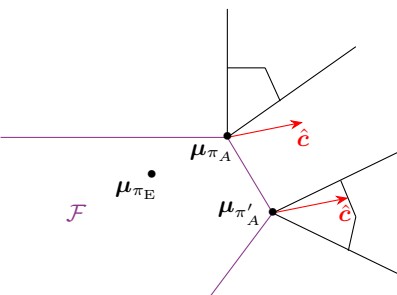

Figure 2: Illustration of IO-AL$_\alpha$.

Figure 2 illustrates IO-AL$_\alpha$ using the same notation as Figure 1. In this scenario, the suboptimal expert lies within the occupancy measure set $\mathcal{F}$ rather than at a vertex. The apprentice policy $\pi_A$ provides a better explanation of the expert's behavior, as it corresponds to the closest vertex, while $\pi_A'$ better aligns with the proxy cost vector $\hat{c}$. The parameter $\alpha$ governs this trade-off: when $\alpha$ increases, the optimization problem selects $\mu_{\pi_A'}$, whereas for values closer to zero, it selects $\mu_{\pi_A}$.

This is our alternative to the AL formalism: instead of identifying the set of vectors $\{c_i\}_{i=1}^{n_c}$ that define $\mathcal{C}_{\text{conv}}$ and choosing an arbitrary cost vector within the inverse-feasible set, we search over a general convex class of cost vectors $\mathcal{C}$ and define an estimate $\hat{c}$ to guide the search. In practice, obtaining information about optimal experts is challenging (Brown et al., 2019; Chen et al., 2021; Wang et al., 2021). Furthermore, given an expert's policy or demonstrations, it is difficult to determine whether it is optimal. Nonetheless, we still aim to leverage suboptimal experts' information and comprehend its actions by solving IO-AL$_\alpha$.

### 3.4 Min-max formulation

We aim to reformulate IO-AL$_\alpha$ as a convex-concave-min-max problem and solve this unconstrained optimization problem using stochastic mirror descent. To this end, we compute its Lagrangian:

$$\mathcal{L}(c, u, \mu) = \alpha \|c - \hat{c}\|_2^2 + \langle \mu_{\pi_E} - \mu, c - T_\gamma^\top u \rangle$$

where $\boldsymbol{\mu} \in \mathbb{R}^{|\mathcal{S}||\mathcal{A}|}$ and $\boldsymbol{\mu} \geq \mathbf{0}$. Observe that $\mathcal{L}(\boldsymbol{c}, \boldsymbol{u}, \boldsymbol{\mu})$ is convex on $(\boldsymbol{c}, \boldsymbol{u})$ and concave on $\boldsymbol{\mu}$. Thus, IO-AL$_\alpha$ is equivalent to the min-max problem

$$\min_{\boldsymbol{c} \in \mathcal{C}, \boldsymbol{u} \in \mathbb{R}^{|\mathcal{S}|}} \max_{\boldsymbol{\mu} \geq \mathbf{0}} \mathcal{L}(\boldsymbol{c}, \boldsymbol{u}, \boldsymbol{\mu}).$$

In our setting, we assume that $\mathcal{C} = \mathbb{B}_1^{|\mathcal{S}||\mathcal{A}|}$, which is not restrictive because we can scale any cost vector to lie within this set. Therefore, we know that $\|\boldsymbol{V}_{\boldsymbol{c}}^\pi\|_\infty \leq 1$ for any policy $\pi \in \Pi_0$ and $\boldsymbol{c} \in \mathcal{C}$ (see Lemma 3 in the Appendix). Hence, we can search for $(\boldsymbol{c}, \boldsymbol{u})$ within the box $\mathbb{B}_1^{|\mathcal{S}||\mathcal{A}|} \times \mathbb{B}_1^{|\mathcal{S}|}$. Moreover, as all feasible solutions for MDP-P$_{\boldsymbol{c}}$ belong to the simplex $\Delta^{|\mathcal{S}||\mathcal{A}|}$, we can restrict the search for $\boldsymbol{\mu}$ to the same simplex.

$$\min_{(\boldsymbol{c}, \boldsymbol{u}) \in \mathbb{B}_1^{|\mathcal{S}||\mathcal{A}|} \times \mathbb{B}_1^{|\mathcal{S}|}} \max_{\boldsymbol{\mu} \in \Delta^{|\mathcal{S}||\mathcal{A}|}} \alpha\|\boldsymbol{c} - \hat{\boldsymbol{c}}\|_2^2 + \langle \boldsymbol{\mu}_{\pi_E} - \boldsymbol{\mu}, \boldsymbol{c} - \boldsymbol{T}_\gamma^\top \boldsymbol{u} \rangle. \tag{RLfD$_\alpha$}$$

Observe that this formulation closely resembles previous min-max formulations of the LfD problem (Kamoutsi et al., 2021). It can be interpreted as a regularized version of this problem, where the search for $\boldsymbol{c}$ is conducted within a general class of cost functions rather than being restricted to a previously specified convex hull.

## 4 Algorithm

Revisiting the assumptions for our problem, we assume that we have access to a generative-model oracle of the expert's occupancy measure $\boldsymbol{\mu}_{\pi_E}$, as well as a generative-model oracle for the MDP's transition law. In this section, we will focus on solving RLfD$_\alpha$ via stochastic mirror descent. Before attempting to solve this problem, we must first define what constitutes a good solution. We define an $\epsilon$-approximate solution as a pair $(\boldsymbol{c}, \boldsymbol{u}), \boldsymbol{\mu}$ such that their duality gap is bounded by $\epsilon > 0$.

**Definition 1** ($\epsilon$-approximate solution). *Given $\epsilon > 0$, an $\epsilon$-approximate solution of RLfD$_\alpha$ is a pair of feasible solutions $((\boldsymbol{c}^\epsilon, \boldsymbol{u}^\epsilon), \boldsymbol{\mu}^\epsilon) \in \left( \mathbb{B}_1^{|\mathcal{S}||\mathcal{A}|} \times \mathbb{B}_1^{|\mathcal{S}|} \right) \times \Delta^{|\mathcal{S}||\mathcal{A}|}$ that satisfy*

$$Gap((\boldsymbol{c}^\epsilon, \boldsymbol{u}^\epsilon), \boldsymbol{\mu}^\epsilon) = \max_{\boldsymbol{\mu}' \in \Delta^{|\mathcal{S}||\mathcal{A}|}} \mathcal{L}((\boldsymbol{c}^\epsilon, \boldsymbol{u}^\epsilon), \boldsymbol{\mu}') - \min_{(\boldsymbol{c}', \boldsymbol{u}') \in \mathbb{B}_1^{|\mathcal{S}||\mathcal{A}|} \times \mathbb{B}_1^{|\mathcal{S}|}} \mathcal{L}((\boldsymbol{c}', \boldsymbol{u}'), \boldsymbol{\mu}^\epsilon) \leq \epsilon.$$

To minimize the duality gap, we require descent and ascent directions. The gradients of $\mathcal{L}((\boldsymbol{c}, \boldsymbol{u}), \boldsymbol{\mu})$ at a given iterate $((\boldsymbol{c}_t, \boldsymbol{u}_t), \boldsymbol{\mu}_t) \in (\mathbb{B}_1^{|\mathcal{S}||\mathcal{A}|} \times \mathbb{B}_1^{|\mathcal{S}|}) \times \Delta^{|\mathcal{S}||\mathcal{A}|}$ are given by

$$g^{(\boldsymbol{c}, \boldsymbol{u})}((\boldsymbol{c}_t, \boldsymbol{u}_t), \boldsymbol{\mu}_t) = \begin{pmatrix} 2\alpha(\boldsymbol{c}_t - \hat{\boldsymbol{c}}) + \boldsymbol{\mu}_{\pi_E} - \boldsymbol{\mu}_t \\ \boldsymbol{T}_\gamma \boldsymbol{\mu}_t - \boldsymbol{T}_\gamma \boldsymbol{\mu}_{\pi_E} \end{pmatrix},$$

$$g^{\boldsymbol{\mu}}((\boldsymbol{c}_t, \boldsymbol{u}_t), \boldsymbol{\mu}_t) = -(-\boldsymbol{c}_t + \boldsymbol{T}_\gamma^\top \boldsymbol{u}_t) = \boldsymbol{c}_t - \boldsymbol{T}_\gamma^\top \boldsymbol{u}_t,$$

where $g^{(\boldsymbol{c}, \boldsymbol{u})}((\boldsymbol{c}_t, \boldsymbol{u}_t), \boldsymbol{\mu}_t) = \nabla_{(\boldsymbol{c}, \boldsymbol{u})} \mathcal{L}((\boldsymbol{c}_t, \boldsymbol{u}_t), \boldsymbol{\mu}_t)$ and $g^{\boldsymbol{\mu}}((\boldsymbol{c}_t, \boldsymbol{u}_t), \boldsymbol{\mu}_t) = -\nabla_{\boldsymbol{\mu}} \mathcal{L}((\boldsymbol{c}_t, \boldsymbol{u}_t), \boldsymbol{\mu}_t)$. Since explicit access to $\boldsymbol{T}_\gamma$ and $\boldsymbol{\mu}_{\pi_E}$ is unavailable, it is necessary to develop gradient estimators that are compatible with oracle-based queries.

**Definition 2** (bounded estimator). *Given the following properties on mean, scale, and variance of an estimator:*

  *i. unbiasedness: $\mathbb{E}[\tilde{g}] = g$.*

  *ii. bounded maximum entry: $\|\tilde{g}\|_\infty \leq z$ with probability 1.*

  *iii. bounded second-moment: $\mathbb{E}[\|\tilde{g}\|^2] \leq v$*

*we call $\tilde{g}$ a $(v, \|\cdot\|)$-bounded estimator if it satisfies (i) and (iii) and a $(z, v, \|\cdot\|_{\Delta^m})$-bounded estimator if it satisfies (i), (ii), (iii) with local norm $\|\cdot\|_{\boldsymbol{y}}$ for all $\boldsymbol{y} \in \Delta^m$.*

With this in mind, define the gradient estimator for the $(\boldsymbol{c}, \boldsymbol{u})$ side through the following procedure

$$\text{sample} \quad (s, a) \sim \frac{1}{|\mathcal{S}||\mathcal{A}|}, \; (s_t, a_t) \sim \boldsymbol{\mu}_t, \; s'_t \sim P(\cdot \mid s_t, a_t), \; (s_E, a_E) \sim \boldsymbol{\mu}_{\pi_E}, \; s'_E \sim P(\cdot \mid s_E, a_E),$$

$$\text{set} \quad \tilde{g}^{(\boldsymbol{c}, \boldsymbol{u})}((\boldsymbol{c}_t, \boldsymbol{u}_t), \boldsymbol{\mu}_t) = \begin{pmatrix} |\mathcal{S}||\mathcal{A}| \cdot 2\alpha \left( c_t(s, a) \boldsymbol{e}^{(s,a)} - \hat{c}(s, a) \boldsymbol{e}^{(s,a)} \right) + \boldsymbol{e}^{(s_E, a_E)} - \boldsymbol{e}^{(s_t, a_t)} \\ \frac{1}{(1-\gamma)} \left( \boldsymbol{e}^{s_t} - \gamma \boldsymbol{e}^{s'_t} - (\boldsymbol{e}^{s_E} - \gamma \boldsymbol{e}^{s'_E}) \right) \end{pmatrix}. \quad (1)$$

In Lemma 1, we show that this estimator is unbiased and provides a bound for its second moment.

**Lemma 1.** *Gradient estimator $\tilde{g}^{(\boldsymbol{c}, \boldsymbol{u})}((\boldsymbol{c}_t, \boldsymbol{u}_t), \boldsymbol{\mu}_t)$ is a $(v^{(\boldsymbol{c}, \boldsymbol{u})}, \|\cdot\|_2)$-bounded estimator, with*

$$v^{(\boldsymbol{c}, \boldsymbol{u})} = 64\alpha^2 \cdot |\mathcal{S}||\mathcal{A}| + \frac{4(1+\gamma^2)}{(1-\gamma)^2} + 8.$$

For the $\boldsymbol{\mu}$ side, define the gradient estimator by

$$\text{sample} \quad (s, a) \sim \frac{1}{|\mathcal{S}||\mathcal{A}|}, \quad s' \sim P(\cdot \mid s, a),$$

$$\text{set} \quad \tilde{g}^{\boldsymbol{\mu}}((\boldsymbol{c}_t, \boldsymbol{u}_t), \boldsymbol{\mu}_t) = |\mathcal{S}||\mathcal{A}| \left( c_t(s, a) \boldsymbol{e}^{(s,a)} - \frac{1}{(1-\gamma)} (u_t(s) \boldsymbol{e}^{(s,a)} - \gamma u_t(s') \boldsymbol{e}^{(s,a)}) \right) \quad (2)$$

As before, we will demonstrate unbiasedness and bound its second moment; however, this time we will also calculate a bound on its maximum entry.

**Lemma 2.** *Gradient estimator $\tilde{g}^{\boldsymbol{\mu}}((\boldsymbol{c}_t, \boldsymbol{u}_t), \boldsymbol{\mu}_t)$ is a $(z^{\boldsymbol{\mu}}, v^{\boldsymbol{\mu}}, \|\cdot\|_2)$-bounded estimator, with*

$$z^{\boldsymbol{\mu}} = \frac{2|\mathcal{S}||\mathcal{A}|}{(1-\gamma)} \text{ and } v^{\boldsymbol{\mu}} = |\mathcal{S}||\mathcal{A}| \left( 2 + \frac{4(1+\gamma^2)}{(1-\gamma)^2} \right).$$

Using these gradient estimators and the bounds established above, we adapt the SMD algorithm originally designed for solving MDPs in Jin & Sidford (2020). Algorithm 1 presents the SMD method for IRL. This algorithm iteratively computes bounded gradient estimators (Lines 3 and 5) by sampling from the occupancy measures and querying the oracle (Lines 2 and 4). The updates are then obtained using mirror descent steps followed by a projection (Lines 6 and 7). After $T$ it-

erations, the algorithm returns the average of the iterates as an $\epsilon$-approximate solution to $\text{RLfD}_\alpha$.

---

**Algorithm 1:** Stochastic Mirror Descent for Inverse Reinforcement Learning

---

**Parameters:** Step-size $\eta^{(\boldsymbol{c},\boldsymbol{u})}$, $\eta^{\boldsymbol{\mu}}$, number of iterations $T$, accuracy level $\epsilon$.

**Input:** State space $\mathcal{S}$, action space $\mathcal{A}$, transition oracle $P$, occupancy measure oracle $\boldsymbol{\mu}_{\pi_E}$, initial state distribution $\boldsymbol{\nu}_0$, discount factor $\gamma$, initial $((\boldsymbol{c}_0, \boldsymbol{u}_0), \boldsymbol{\mu}_0) \in \mathbb{B}_1^{|\mathcal{S}||\mathcal{A}|} \times \Delta^{|\mathcal{S}||\mathcal{A}|}$.

**Output:** An expected $\epsilon$-approximate solution $((\boldsymbol{c}^\epsilon, \boldsymbol{u}^\epsilon), \boldsymbol{\mu}^\epsilon)$ for $\text{RLfD}_\alpha$.

1 **for** $t \leftarrow 0$ **to** $T-1$ **do**

$\quad$ /* $(\boldsymbol{c}, \boldsymbol{u})$ gradient estimation $\qquad\qquad\qquad\qquad\qquad\qquad\qquad\qquad\qquad\qquad$ */

2 $\quad$ Sample $(s_t, a_t) \sim \boldsymbol{\mu}_t$, $s_t' \sim P(\cdot \mid s_t, a_t)$, $(s_E, a_E) \sim \boldsymbol{\mu}_{\pi_E}$, $s_E' \sim P(\cdot \mid s_E, a_E)$

3 $\quad$ Compute:

$$\tilde{g}^{(\boldsymbol{c},\boldsymbol{u})}((\boldsymbol{c}_t, \boldsymbol{u}_t), \boldsymbol{\mu}_t) = \begin{pmatrix} 2\alpha(\boldsymbol{c}_t - \hat{\boldsymbol{c}}) + \boldsymbol{\mu}_{\pi_E} - \boldsymbol{\mu}_t \\ \frac{1}{(1-\gamma)}\left(\boldsymbol{e}^{s_t} - \gamma \boldsymbol{e}^{s_t'} - (\boldsymbol{e}^{s_E} - \gamma \boldsymbol{e}^{s_E'})\right) \end{pmatrix}$$

$\quad$ /* $\boldsymbol{\mu}$ gradient estimation $\qquad\qquad\qquad\qquad\qquad\qquad\qquad\qquad\qquad\qquad\qquad$ */

4 $\quad$ Sample $(s, a) \sim \frac{1}{|\mathcal{S}||\mathcal{A}|}$, $s' \sim P(\cdot \mid s, a)$

5 $\quad$ Compute:

$$\tilde{g}^{\boldsymbol{\mu}}((\boldsymbol{c}_t, \boldsymbol{u}_t), \boldsymbol{\mu}_t) = |\mathcal{S}||\mathcal{A}|\left(c_t(s,a)\boldsymbol{e}^{(s,a)} - \frac{1}{(1-\gamma)}(u_t(s)\boldsymbol{e}^{(s,a)} - \gamma u_t(s')\boldsymbol{e}^{(s,a)})\right)$$

$\quad$ /* Mirror descent steps $\qquad\qquad\qquad\qquad\qquad\qquad\qquad\qquad\qquad\qquad\qquad\quad$ */

6 $\quad$ $(\boldsymbol{c}_t, \boldsymbol{u}_t) \leftarrow \Pi_{\mathbb{B}_1^{|\mathcal{S}||\mathcal{A}|} \times \mathbb{B}_1^{|\mathcal{S}|}}\left((\boldsymbol{c}_{t-1}, \boldsymbol{u}_{t-1}) - \eta^{(\boldsymbol{c},\boldsymbol{u})}\tilde{g}^{(\boldsymbol{c},\boldsymbol{u})}((\boldsymbol{c}_{t-1}, \boldsymbol{u}_{t-1}), \boldsymbol{\mu}_{t-1})\right)$

7 $\quad$ $\boldsymbol{\mu}_t \leftarrow \Pi_{\Delta^{|\mathcal{S}||\mathcal{A}|}}\left(\boldsymbol{\mu}_{t-1} \circ \exp(-\eta^{\boldsymbol{\mu}}\tilde{g}^{\boldsymbol{\mu}}((\boldsymbol{c}_{t-1}, \boldsymbol{u}_{t-1}), \boldsymbol{\mu}_{t-1}))\right)$

8 **return** $((\boldsymbol{c}^\epsilon, \boldsymbol{u}^\epsilon), \boldsymbol{\mu}^\epsilon) \leftarrow \frac{1}{T}\sum_{t=1}^{T}((\boldsymbol{c}_t, \boldsymbol{u}_t), \boldsymbol{\mu}_t)$

---

**Theorem 2.** *Given $\epsilon \in (0, 1)$, Algorithm 1 with step-size*

$$\eta^{(\boldsymbol{c},\boldsymbol{u})} = \frac{\epsilon}{4v^{(\boldsymbol{c},\boldsymbol{u})}}, \quad \eta^{\boldsymbol{\mu}} = \frac{\epsilon}{4v^{\boldsymbol{\mu}}},$$

*and gradient estimators defined in equations 1 and 2 finds an expected $\epsilon$-approximate solution*

$$\mathbb{E}\left[Gap((\boldsymbol{c}^\epsilon, \boldsymbol{u}^\epsilon), \boldsymbol{\mu}^\epsilon)\right] \leq \epsilon,$$

*within any iteration number*

$$T \geq \max\left\{\mathcal{O}\left(\frac{\alpha^2|\mathcal{S}|^3|\mathcal{A}|^2}{\epsilon^2}\right), \mathcal{O}\left(\frac{|\mathcal{S}||\mathcal{A}|\log(|\mathcal{S}||\mathcal{A}|)}{\epsilon^2}\right)\right\}.$$

The number of iterations scales quadratically with the number of actions and cubically with the number of states. Theoretically, the number of iterations required depends on the parameter $\alpha \in \mathbb{R}_{\geq 0}$. When $\alpha = 0$, the number of iterations decreases significantly as the $|\mathcal{S}|^3$ and $|\mathcal{A}|^2$ terms vanish from the initial expression. This suggests that introducing the regularization $\alpha\|\boldsymbol{c} - \hat{\boldsymbol{c}}\|_2^2$ increases the complexity of the problem. Nevertheless, we will see in the next section that the regularization term helps to guide the search to uncover the true cost function.

**Proposition 4.** *Let $((\boldsymbol{c}^\epsilon, \boldsymbol{u}^\epsilon)\boldsymbol{\mu}^\epsilon)$ be an expected $\epsilon$-approximate solution for $\text{RLfD}_\alpha$, such that $\boldsymbol{\mu}^\epsilon$ induces a policy $\pi_{\boldsymbol{\mu}^\epsilon} \in \Pi_0$ defined by $\pi_{\boldsymbol{\mu}^\epsilon}(a|s) = \frac{\boldsymbol{\mu}(a,s)}{\sum_{a'}\boldsymbol{\mu}(s,a)}$. It then holds that*

$$\mathbb{E}\left[\alpha\|\boldsymbol{c}^\epsilon - \hat{\boldsymbol{c}}\|_2^2 + \rho_{\boldsymbol{c}^\epsilon}(\pi_E) - \rho_{\boldsymbol{c}^\epsilon}(\pi_A)\right] \leq \epsilon + \alpha\|\boldsymbol{c}_A - \hat{\boldsymbol{c}}\|_2^2 + \rho_{\boldsymbol{c}_A}(\pi_E) - \rho_{\boldsymbol{c}_A}^*,$$

*where $((\boldsymbol{c}_A, \boldsymbol{u}_A), \boldsymbol{\mu}_A)$ denotes the optimal solution for $\text{RLfD}_\alpha$ and $\pi_A$ is the policy induced by $\boldsymbol{\mu}_A$.*

Proposition 4 establishes a connection between expected $\epsilon$-approximate solutions for $\text{RLfD}_\alpha$ and its optimal solution. Specifically, it shows that an expected $\epsilon$-approximate solution achieves an objective value that is at most $\epsilon$ worse than that of the optimal solution. Note that we assume that $\boldsymbol{\mu}^\epsilon$ induces a policy, which is not always the case for an arbitrary $\boldsymbol{\mu} \in \Delta^{|\mathcal{S}||\mathcal{A}|}$.

# 5 Numerical experiments

We use a standard $H \times W$ Gridworld environment (Figure 3(a)), where each cell is a unique state. Obstacles (shown in red) incur a cost of 1, terminal cells (shown in green) incur a cost of $-1$, and all other cells (shown in white) incur a cost of 0. The action set is $\{\text{up}, \text{down}, \text{left}, \text{right}\}$, but a "wind" introduces a 20% chance of drifting left or otherwise altering the intended move. If the resulting move is out of bounds, the agent remains in the same cell. The discount factor is 0.7, and initial states are chosen uniformly. We solve the corresponding MDP-P$_{\boldsymbol{c}}$ with a linear solver to obtain the optimal occupancy measure (Figure 3(b)). To construct a suboptimal expert, we terminate the solver early and use the resulting $\boldsymbol{\mu}$ as the expert's occupancy measure (Figure 3(c)). To visualize the policies induced by these occupancy measures, we extract the most likely action at each state by computing $\arg\max_{a \in \mathcal{A}} \boldsymbol{\mu}(s, a)$ and display it in the corresponding cell.

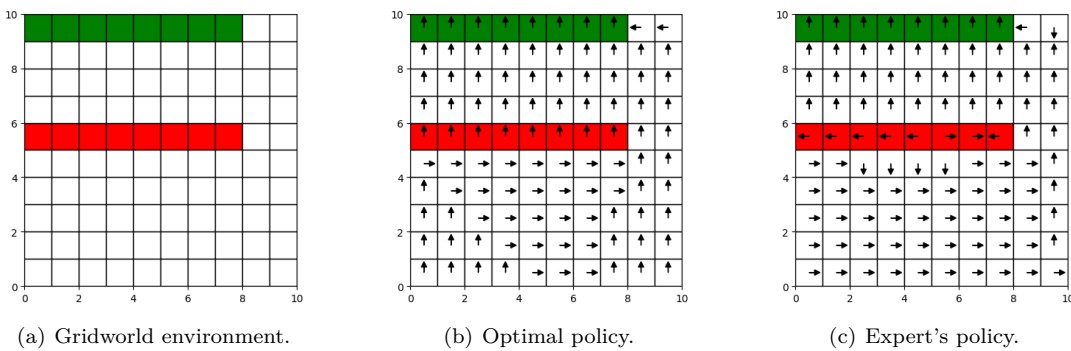

(a) Gridworld environment.   (b) Optimal policy.   (c) Expert's policy.

Figure 3: Illustration of the Gridworld environment, the optimal policy, and the expert's policy.

## 5.1 Implementation details

We executed Algorithm 1 $N$ times with randomly generated initial values for $T$ iterations and averaged the resulting outputs. With reproducibility in mind, we provide a Python package (see supplementary material) with a framework that can handle general discounted Markov decision processes.

## 5.2 Regularization effect

We study the effect of the regularization term $\alpha\|\mathbf{c} - \hat{\mathbf{c}}\|_2^2$ in solving problem RLfD$_\alpha$ via SMD. To this end, we define a cost vector $\hat{\boldsymbol{c}}$ that is zero everywhere except for a randomly selected subset of obstacle and goal states. This choice was made based on practical considerations, as in most real-world scenarios, we rarely have access to a highly accurate estimate of the cost vector. Specifically, for obstacles, $\hat{\boldsymbol{c}}$ is set to 1 for a randomly selected 50% of the obstacle states, while for goal states, $\hat{\mathbf{c}}$ is set to $-1$ for a randomly chosen 50% of them. Moreover, regarding the algorithm's parameters, we chose $N = 20$, $T = 10^5$, and both step-sizes as $10^{-2}$.

Figure 4 depicts the learned cost vectors for each action $\{\text{up}, \text{down}, \text{left}, \text{right}\}$ under varying levels of regularization $\alpha$. As $\alpha$ increases, the cost vectors display more white regions, indicating near-zero cost values, and more accurately highlight the main obstacles. In turn, this leads to a better approximation of the true cost structure. However, when the regularization is too strong, it may overly penalize costs associated with obstacle areas that are less frequently demonstrated, thereby ignoring parts of the environment that do not appear in the demonstration data. Consequently, selecting an appropriate $\alpha$ is crucial to achieve a cost vector that balances fidelity to the true environment and robustness in identifying cost structures that are not present in the estimate $\hat{\boldsymbol{c}}$.

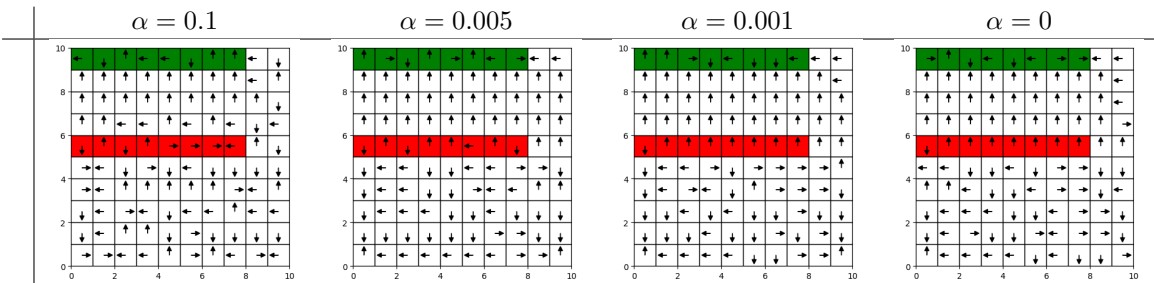

Figure 4: Effect of the regularization on the cost vector.

Figure 5 displays the apprentice policy obtained using SMD for various values of $\alpha$. Notably, when $\alpha$ is set to 0.005 or 0.001, the apprentice policy closely resembles the one derived by exclusively weighting the expert's policy (i.e., $\alpha = 0$). This observation is significant, as the previous analysis demonstrated that incorporating regularization yields a cost vector for the apprentice policies that aligns more closely with the true cost vector.

Figure 5: Effect of regularization on the apprentice policy.

### 5.3 Convergence

In the same experimental setting, we examined the convergence of the solutions up to iteration $t < T$ by computing the $L^1$-norm of the difference between the solutions at iterations $t$ and $t-1$. We plotted this norm for each of the $\alpha$ values considered in the previous experiments. According to this sense of convergence, all $\alpha$ values exhibit comparable convergence rates for $\boldsymbol{u}$ and $\boldsymbol{\mu}$. However, for $\boldsymbol{c}$, the convergence rate at $\alpha = 0.1$ is faster than that observed for smaller values of $\alpha$. This behavior aligns with the increased penalty imposed for deviating from $\hat{\boldsymbol{c}}$ under stronger regularization, thereby accelerating convergence for $\boldsymbol{c}$.

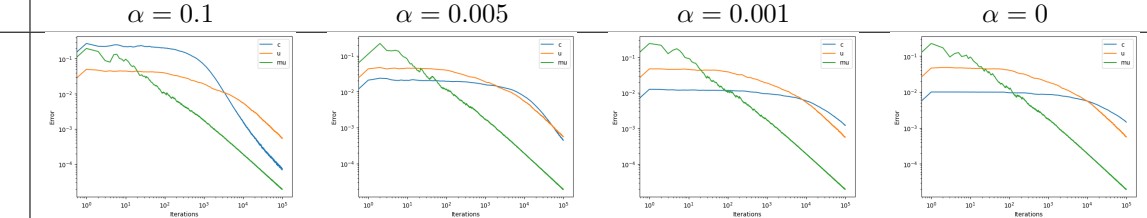

Figure 6: Effect of regularization on the difference of solutions found between iterations.

However, it is important to note that Theorem 2 characterizes convergence in terms of the duality gap. In Figure 7, we compute the duality gap of the solution every 25 iterations, using IPOPT (Wächter & Biegler, 2006). The algorithm parameters remain unchanged, except that, due to computational constraints, we reduce the grid size to $6 \times 6$ and limit the execution to $T = 10^4$ iterations. As expected from the results of Theorem 2, stronger regularization leads to slower convergence of the duality gap, a trend illustrated in this figure. While we did not use the theoretically prescribed step sizes due to their small magnitude, the figure still offers valuable insights into how regularization affects the convergence behavior.

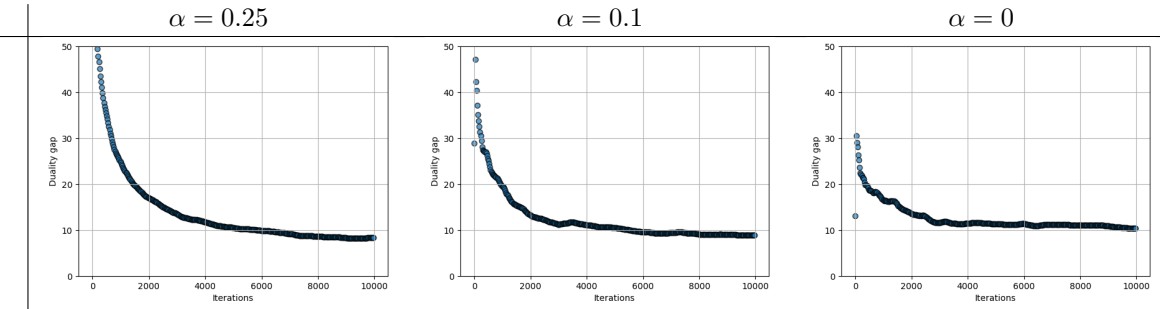

Figure 7: Duality gap convergence.

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

# A  Appendix

## A.1  MDPs

**Lemma 3.** *Given a DMDP with discount factor $\gamma \in (0, 1)$, then the value function satisfies*

$$V_{\boldsymbol{c}}^{\pi}(s) \leq \|\boldsymbol{c}\|_{\infty} = 1$$

*for any policy $\pi \in \Pi_0$ and any state $s \in \mathcal{S}$.*

*Proof.* Given a policy $\pi \in \Pi_0$ and a state $s \in \mathcal{S}$ the value function

$$
\begin{aligned}
V_{\boldsymbol{c}}^{\pi}(s) &= (1 - \gamma)\mathbb{E}_s^{\pi}\left[\sum_{t=0}^{\infty} \gamma^t c(s_t, a_t)\right] \\
&\leq (1 - \gamma)\mathbb{E}_s^{\pi}\left[\sum_{t=0}^{\infty} \gamma^t \|\boldsymbol{c}\|_{\infty}\right] \\
&= (1 - \gamma)\frac{1}{(1 - \gamma)}\|\boldsymbol{c}\|_{\infty} \\
&= 1.
\end{aligned}
$$

$\square$

### A.2 IO, LfD, and AL

**Theorem 1.** *Suppose that $\boldsymbol{\mu}_{\pi_E}$ is an optimal solution for equation MDP-$P_{\boldsymbol{c}}$ where $\boldsymbol{c} \in \mathcal{C}$. Then the following equality holds:*

$$\Theta^{inv}(\boldsymbol{\mu}_{\pi_E}) = \Pi_1\left(\arg\max_{(\boldsymbol{c},\boldsymbol{u})}\{\langle \boldsymbol{\mu}_{\pi_E}, \boldsymbol{T}_\gamma^\top \boldsymbol{u} - \boldsymbol{c}\rangle \mid \boldsymbol{c} \in \mathcal{C}, \; \boldsymbol{c} - \boldsymbol{T}_\gamma^\top \boldsymbol{u} \geq \boldsymbol{0}\}\right)$$

*where $\Pi_1$ denotes the projection in the first component.*

*Proof.* Note that

$$\arg\max_{(\boldsymbol{c},\boldsymbol{u})}\{\langle \boldsymbol{\mu}_{\pi_E}, \boldsymbol{T}_\gamma^\top \boldsymbol{u} - \boldsymbol{c}\rangle \mid \boldsymbol{c} \in \mathcal{C}, \; \boldsymbol{c} - \boldsymbol{T}_\gamma^\top \boldsymbol{u} \geq \boldsymbol{0}\}$$
$$= \arg\min_{(\boldsymbol{c},\boldsymbol{u})}\{\langle \boldsymbol{\mu}_{\pi_E}, \boldsymbol{c} - \boldsymbol{T}_\gamma^\top \boldsymbol{u}\rangle \mid \boldsymbol{c} \in \mathcal{C}, \; \boldsymbol{c} - \boldsymbol{T}_\gamma^\top \boldsymbol{u} \geq \boldsymbol{0}\},$$

thus, we will prove that

$$\Theta^{inv}(\boldsymbol{\mu}_{\pi_E}) = \Pi_1\left(\arg\min_{(\boldsymbol{c},\boldsymbol{u})}\{\langle \boldsymbol{\mu}_{\pi_E}, \boldsymbol{c} - \boldsymbol{T}_\gamma^\top \boldsymbol{u}\rangle \mid \boldsymbol{c} \in \mathcal{C}, \; \boldsymbol{c} - \boldsymbol{T}_\gamma^\top \boldsymbol{u} \geq \boldsymbol{0}\}\right).$$

If $\boldsymbol{\mu}_{\pi_E} \in \mathcal{F}$, then $\boldsymbol{\mu}_{\pi_E} \geq \boldsymbol{0}$. Using the restriction $\boldsymbol{c} - \boldsymbol{T}_\gamma^\top \boldsymbol{u} \geq \boldsymbol{0}$, we have that

$$\min_{(\boldsymbol{c},\boldsymbol{u})}\{\langle \boldsymbol{\mu}_{\pi_E}, \boldsymbol{c} - \boldsymbol{T}_\gamma^\top \boldsymbol{u}\rangle \mid \boldsymbol{c} \in \mathcal{C}, \; \boldsymbol{c} - \boldsymbol{T}_\gamma^\top \boldsymbol{u} \geq \boldsymbol{0}\} \geq 0.$$

If $\boldsymbol{c}_0 \in \Theta^{inv}(\boldsymbol{\mu}_{\pi_E})$, then there exists $\boldsymbol{u}_0 \in \mathbb{R}^{|\mathcal{S}|}$ that satisfies $\boldsymbol{c}_0 - \boldsymbol{T}_\gamma^\top \boldsymbol{u}_0 \geq \boldsymbol{0}$ and $\langle \boldsymbol{\mu}_{\pi_E}, \boldsymbol{c}_0 - \boldsymbol{T}_\gamma^\top \boldsymbol{u}_0\rangle = 0$. This implies that

$$(\boldsymbol{c}_0, \boldsymbol{u}_0) \in \arg\min_{(\boldsymbol{c},\boldsymbol{u})}\{\langle \boldsymbol{\mu}_{\pi_E}, \boldsymbol{c} - \boldsymbol{T}_\gamma^\top \boldsymbol{u}\rangle \mid \boldsymbol{c} \in \mathcal{C}, \; \boldsymbol{c} - \boldsymbol{T}_\gamma^\top \boldsymbol{u} \geq \boldsymbol{0}\}.$$

On the other hand, if $\boldsymbol{c}^* \in \Pi_1\left(\arg\min_{(\boldsymbol{c},\boldsymbol{u})}\{\langle \boldsymbol{\mu}_{\pi_E}, \boldsymbol{c} - \boldsymbol{T}_\gamma^\top \boldsymbol{u}\rangle \mid \boldsymbol{c} \in \mathcal{C}, \; \boldsymbol{c} - \boldsymbol{T}_\gamma^\top \boldsymbol{u} \geq \boldsymbol{0}\}\right)$, then $\boldsymbol{c}^* \in \mathcal{C}$ and there exists $\boldsymbol{u}^*$ such that

$$\boldsymbol{c}^* - \boldsymbol{T}_\gamma^\top \boldsymbol{u}^* \geq \boldsymbol{0} \text{ and } \langle \boldsymbol{\mu}_{\pi_E}, \boldsymbol{c}^* - \boldsymbol{T}_\gamma^\top \boldsymbol{u}^*\rangle \leq \langle \boldsymbol{\mu}_{\pi_E}, \boldsymbol{c} - \boldsymbol{T}_\gamma^\top \boldsymbol{u}\rangle,$$

for every pair $(\boldsymbol{c}, \boldsymbol{u})$ such that $\boldsymbol{c} \in \mathcal{C}$ and $\boldsymbol{c} - \boldsymbol{T}_\gamma^\top \boldsymbol{u} \geq \boldsymbol{0}$. In particular, this is true for $(\hat{\boldsymbol{c}}, \hat{\boldsymbol{u}})$ where $\boldsymbol{\mu}_{\pi_E}$ is optimal for MDP-$P_{\hat{\boldsymbol{c}}}$ and $\hat{\boldsymbol{u}}$ is its dual optimal. Therefore, by complementary slackness we have that

$$0 \leq \langle \boldsymbol{\mu}_{\pi_E}, \boldsymbol{c}^* - \boldsymbol{T}_\gamma^\top \boldsymbol{u}^*\rangle \leq \langle \boldsymbol{\mu}_{\pi_E}, \hat{\boldsymbol{c}} - \boldsymbol{T}_\gamma^\top \hat{\boldsymbol{u}}\rangle = 0,$$

and we conclude that $\boldsymbol{c}^* \in \Theta^{inv}(\boldsymbol{\mu}_{\pi_E})$. $\qquad\square$

**Proposition 3.** *A pair $(\boldsymbol{c}_A, \boldsymbol{u}_A)$ is optimal for IO-$AL_\alpha$ if and only if the apprentice policy $\pi_A$ is optimal for $RL_{\boldsymbol{c}_A}$ and $\boldsymbol{u}_A = \boldsymbol{V}_{\boldsymbol{c}_A}^*$. Furthermore, the optimal value corresponds to $\alpha\|\boldsymbol{c}_A - \hat{\boldsymbol{c}}\|_2^2 + \rho_{\boldsymbol{c}_A}(\pi_E) - \rho_{\boldsymbol{c}_A}^*$.*

*Proof.* Observe that IO-$AL_\alpha$ is a convex optimization problem. Therefore, a pair $(\boldsymbol{c}_A, \boldsymbol{u}_A)$ is optimal for IO-$AL_\alpha$ if and only if the KKT conditions are satisfied. In particular, it satisfies complementary slackness $\langle \boldsymbol{\mu}_{\pi_A}, \boldsymbol{c}_A - \boldsymbol{T}_\gamma^\top \boldsymbol{u}_A\rangle = 0$ and the stationarity condition $\boldsymbol{T}_\gamma \boldsymbol{\mu}_{\pi_A} = \boldsymbol{T}_\gamma \boldsymbol{\mu}_{\pi_E} = \boldsymbol{\nu}_0$, which implies that $\boldsymbol{\mu}_{\pi_A} \in \mathcal{F}$. Then, by Proposition 2 $\pi_A$ is optimal for RL$_{\boldsymbol{c}_A}$ with $\boldsymbol{u}_A = \boldsymbol{V}_{\boldsymbol{c}_A}^*$. Moreover, if $(\boldsymbol{c}_A, \boldsymbol{u}_A)$ are optimal we have that

$$\alpha\|\boldsymbol{c}_A - \hat{\boldsymbol{c}}\|_2^2 + \langle \boldsymbol{\mu}_{\pi_E}, \boldsymbol{c}_A - \boldsymbol{T}_\gamma^\top \boldsymbol{u}_A\rangle = \alpha\|\boldsymbol{c}_A - \hat{\boldsymbol{c}}\|_2^2 + \langle \boldsymbol{\mu}_{\pi_E}, \boldsymbol{c}_A\rangle - \langle \boldsymbol{\mu}_{\pi_E}, \boldsymbol{T}_\gamma^\top \boldsymbol{u}_A\rangle$$
$$= \alpha\|\boldsymbol{c}_A - \hat{\boldsymbol{c}}\|_2^2 + \rho_{\boldsymbol{c}_A}(\pi_E) - \langle \boldsymbol{\nu}_0, \boldsymbol{u}_A\rangle$$
$$= \alpha\|\boldsymbol{c}_A - \hat{\boldsymbol{c}}\|_2^2 + \rho_{\boldsymbol{c}_A}(\pi_E) - \rho_{\boldsymbol{c}_A}^*.$$

$\qquad\square$

### A.3 Gradient estimation

**Lemma 1.** *Gradient estimator* $\tilde{g}^{(\boldsymbol{c},\boldsymbol{u})}((\boldsymbol{c}_t, \boldsymbol{u}_t), \boldsymbol{\mu}_t)$ *is a* $(v^{(\boldsymbol{c},\boldsymbol{u})}, \|\cdot\|_2)$*-bounded estimator, with*

$$v^{(\boldsymbol{c},\boldsymbol{u})} = 64\alpha^2 \cdot |\mathcal{S}||\mathcal{A}| + \frac{4(1+\gamma^2)}{(1-\gamma)^2} + 8.$$

*Proof.* The unbiasedness follows from the following observations

$$\sum_{s,a} \frac{|\mathcal{S}||\mathcal{A}|}{|\mathcal{S}||\mathcal{A}|} \cdot 2\alpha \left( c_t(s,a)\boldsymbol{e}^{(s,a)} - \hat{c}(s,a)\boldsymbol{e}^{(s,a)} \right)$$

$$+ \sum_{s_E, a_E} \mu_{\pi_E}(s_E, a_E)\boldsymbol{e}^{(s_E, a_E)} - \sum_{s_t, a_t} \mu_t(s_t, a_t)\boldsymbol{e}^{(s_t, a_t)}$$

$$= 2\alpha(\boldsymbol{c}_t - \hat{\boldsymbol{c}}) + \boldsymbol{\mu}_{\pi_E} - \boldsymbol{\mu}_t$$

and

$$\frac{1}{(1-\gamma)} \sum_{s'_t, a_t, s_t} \mu_t(s_t, a_t) P(s'_t \mid s_t, a_t)(\boldsymbol{e}^{(s_t)} - \gamma\boldsymbol{e}^{(s'_t)})$$

$$= \frac{1}{(1-\gamma)} \left( \sum_{s_t, a_t} \mu_t(s_t, a_t)\boldsymbol{e}^{(s_t)} - \gamma \sum_{s'_t, a_t, s_t} \mu_t(s_t, a_t) P(s'_t \mid s_t, a_t)\boldsymbol{e}^{(s'_t)} \right)$$

$$= \frac{1}{(1-\gamma)}(\boldsymbol{B}\boldsymbol{\mu} - \gamma\boldsymbol{P}\boldsymbol{\mu})$$

$$= \boldsymbol{T}_\gamma \boldsymbol{\mu}.$$

Thus, we obtain

$$\mathbb{E}\left[\tilde{g}^{(\boldsymbol{c},\boldsymbol{u})}((\boldsymbol{c},\boldsymbol{u}),\boldsymbol{\mu})\right] = \begin{pmatrix} 2\alpha(\boldsymbol{c}_t - \hat{\boldsymbol{c}}) + \boldsymbol{\mu}_{\pi_E} - \boldsymbol{\mu}_t \\ \boldsymbol{T}_\gamma\boldsymbol{\mu} - \boldsymbol{T}_\gamma\boldsymbol{\mu}_{\pi_E} \end{pmatrix}.$$

For proving the bound on the second-moment, note that

$$\mathbb{E}\left[\left\| |\mathcal{S}||\mathcal{A}| \cdot 2\alpha \left( c_t(s,a)\boldsymbol{e}^{(s,a)} - \hat{c}(s,a)\boldsymbol{e}^{(s,a)} \right) + \boldsymbol{e}^{(s_E, a_E)} - \boldsymbol{e}^{(s_t, a_t)} \right\|_2^2 \right]$$

$$\leq 2\mathbb{E}\left[ 4|\mathcal{S}||\mathcal{A}|\alpha^2 \left\| c_t(s,a)\boldsymbol{e}^{(s,a)} - \hat{c}(s,a)\boldsymbol{e}^{(s,a)} \right\|_2^2 + \left\| \boldsymbol{e}^{(s_E, a_E)} - \boldsymbol{e}^{(s_t, a_t)} \right\|_2^2 \right]$$

$$\leq 2\mathbb{E}\left[ 4|\mathcal{S}||\mathcal{A}|\alpha^2 \cdot 4 + 2 \right]$$

$$= 32|\mathcal{S}||\mathcal{A}|\alpha^2 + 4,$$

and

$$\mathbb{E}\left[\left\| \frac{1}{(1-\gamma)} \left( \boldsymbol{e}^{(s_t)} - \gamma\boldsymbol{e}^{(s'_t)} - (\boldsymbol{e}^{(s_E)} - \gamma\boldsymbol{e}^{(s'_E)}) \right) \right\|_2^2 \right] \leq \mathbb{E}\left[ \frac{2(1+\gamma^2)}{(1-\gamma)^2} \right]$$

$$= \frac{2(1+\gamma^2)}{(1-\gamma)^2}.$$

Hence, we can provide the bound

$$\mathbb{E}\left[\left\| \tilde{g}^{(\boldsymbol{c},\boldsymbol{u})}((\boldsymbol{c},\boldsymbol{u}),\boldsymbol{\mu}) \right\|_2^2 \right] \overset{(i)}{\leq} 2\left[ 32|\mathcal{S}||\mathcal{A}|\alpha^2 + 4 + \frac{2(1+\gamma^2)}{(1-\gamma)^2} \right]$$

$$= 64\alpha^2 \cdot |\mathcal{S}||\mathcal{A}| + \frac{4(1+\gamma^2)}{(1-\gamma)^2} + 8.$$

where in $(i)$ we used $\|\boldsymbol{x} + \boldsymbol{y}\|^2 \leq 2[\|\boldsymbol{x}\|^2 + \|\boldsymbol{y}\|^2]$. $\qquad\square$

**Lemma 2.** *Gradient estimator* $\tilde{g}^{\boldsymbol{\mu}}((\boldsymbol{c}_t, \boldsymbol{u}_t), \boldsymbol{\mu}_t)$ *is a* $(z^{\boldsymbol{\mu}}, v^{\boldsymbol{\mu}}, \|\cdot\|_2)$*-bounded estimator, with*

$$z^{\boldsymbol{\mu}} = \frac{2|\mathcal{S}||\mathcal{A}|}{(1-\gamma)} \text{ and } v^{\boldsymbol{\mu}} = |\mathcal{S}||\mathcal{A}| \left(2 + \frac{4(1+\gamma^2)}{(1-\gamma)^2}\right).$$

*Proof.* The unbiasedness follows from

$$\mathbb{E}[\tilde{g}^{\boldsymbol{\mu}}((\boldsymbol{c}_t, \boldsymbol{u}_t), \boldsymbol{\mu}_t)] = \sum_{s,a,s'} \frac{1}{|\mathcal{S}||\mathcal{A}|} P(s' \mid s, a) \cdot |\mathcal{S}||\mathcal{A}| \left(c_t(s,a)\boldsymbol{e}^{(s,a)} - \frac{1}{(1-\gamma)}(u_t(s)\boldsymbol{e}^{(s,a)} - \gamma u_t(s')\boldsymbol{e}^{(s,a)})\right)$$

$$= \sum_{s,a} c_t(s,a)\boldsymbol{e}^{(s,a)} - \frac{1}{(1-\gamma)} \left(\sum_{s,a} u_t(s)\boldsymbol{e}^{(s,a)} - \gamma \sum_{s,a} \left(\sum_{s'} P(s' \mid s, a)u_t(s')\right) \boldsymbol{e}^{(s,a)}\right)$$

$$= \boldsymbol{c}_t - \frac{1}{(1-\gamma)} \left(\boldsymbol{B}^\top \boldsymbol{u} - \gamma \boldsymbol{P}^\top \boldsymbol{u}\right)$$

$$= \boldsymbol{c}_t - \boldsymbol{T}_\gamma^\top \boldsymbol{u}.$$

For the bound on the maximum entry observe that

$$\|\tilde{g}^{\boldsymbol{\mu}}((\boldsymbol{c}_t, \boldsymbol{u}_t), \boldsymbol{\mu}_t)\|_\infty = |\mathcal{S}||\mathcal{A}| \max_{s,a,s'} \left\{\left|c_t(s,a) - \frac{u_t(s) - \gamma u_t(s')}{(1-\gamma)}\right|\right\}$$

$$\leq |\mathcal{S}||\mathcal{A}| \left(\|\boldsymbol{c}_t\|_\infty + \|\boldsymbol{u}_t\|_\infty \frac{(1+\gamma)}{(1-\gamma)}\right)$$

$$= \frac{2|\mathcal{S}||\mathcal{A}|}{(1-\gamma)}.$$

Finally, the bound on the second-moment can be obtained by:

$$\mathbb{E}\left[\|\tilde{g}^{(\boldsymbol{\mu})}((\boldsymbol{c}_t, \boldsymbol{u}_t), \boldsymbol{\mu}_t)\|_{\boldsymbol{\mu}'}^2\right] \overset{(i)}{\leq} |\mathcal{S}|^2|\mathcal{A}|^2 \cdot \mathbb{E}\left[2\|c_t(s,a)\boldsymbol{e}^{(s,a)}\|_{\boldsymbol{\mu}'}^2 + \frac{4}{(1-\gamma)^2}(\|u_t(s)\boldsymbol{e}^{(s,a)}\|_{\boldsymbol{\mu}'}^2 + \|\gamma u_t(s')\boldsymbol{e}^{(s,a)}\|_{\boldsymbol{\mu}'}^2)\right]$$

$$= |\mathcal{S}|^2|\mathcal{A}|^2 \cdot \mathbb{E}\left[\mu'(s,a)\left(2(c_t(s,a))^2 + \frac{4}{(1-\gamma)^2}(u_t(s))^2 + \frac{4\gamma^2}{(1-\gamma)^2}(u_t(s'))^2\right)\right]$$

$$= |\mathcal{S}||\mathcal{A}|\left[\sum_{s,a} \mu'(s,a)\left(2(c_t(s,a))^2 + \frac{4}{(1-\gamma)^2}(u_t(s))^2\right)\right.$$

$$\left. + \sum_{s',s,a} \mu'(s,a)P(s' \mid s, a)\frac{4\gamma^2}{(1-\gamma)^2}(u_t(s'))^2\right]$$

$$\overset{(ii)}{\leq} |\mathcal{S}||\mathcal{A}|\left[\left(2 + \frac{4}{(1-\gamma)^2}\right)\sum_{s,a} \mu'(s,a) + \frac{4\gamma^2}{(1-\gamma)^2}\sum_{s',s,a} \mu'(s,a)P(s' \mid s, a)\right]$$

$$= |\mathcal{S}||\mathcal{A}|\left(2 + \frac{4(1+\gamma^2)}{(1-\gamma)^2}\right),$$

where we used $\|\boldsymbol{x} + \boldsymbol{y}\|^2 \leq 2[\|\boldsymbol{x}\|^2 + \|\boldsymbol{y}\|^2]$ two times for $(i)$ and that $(\boldsymbol{c}_t, \boldsymbol{u}_t) \in \mathbb{B}_1^{|\mathcal{S}||\mathcal{A}|} \times \mathbb{B}_1^{|\mathcal{S}|}$ for $(ii)$. $\qquad\square$

### A.4  Algorithm convergence

We will follow Jin & Sidford (2020) and show how their results accommodates to our problem.

**Definition 3** ($\ell_\infty$-$\ell_1$ convex-concave min-max problem). *Let* $f : \mathbb{R}^n \times \mathbb{R}^m \to \mathbb{R}$ *be a differentiable function that is convex in* $\boldsymbol{x} \in \mathbb{R}^n$ *and concave in* $\boldsymbol{y} \in \mathbb{R}^m$. *We define the* $\ell_\infty$-$\ell_1$ *convex-concave min-max problem as*

$$\min_{\boldsymbol{x} \in \mathbb{B}_b^n} \max_{\boldsymbol{y} \in \Delta^m} f(\boldsymbol{x}, \boldsymbol{y}).$$

Furthermore, define the operator

$$G(\boldsymbol{z}) = G(\boldsymbol{x}, \boldsymbol{y}) = [\nabla_{\boldsymbol{x}} f(\boldsymbol{x}, \boldsymbol{y}), -\nabla_{\boldsymbol{y}} f(\boldsymbol{x}, \boldsymbol{y})] = [g^{\boldsymbol{x}}(\boldsymbol{x}, \boldsymbol{y}), g^{\boldsymbol{y}}(\boldsymbol{x}, \boldsymbol{y})].$$

**Lemma 3** (cf. Appendix A.1 in Carmon et al. (2019)). *For every $\boldsymbol{z}_1, ..., \boldsymbol{z}_K \in \mathcal{Z} = \mathbb{B}_b^n \times \Delta^m$ it holds that*

$$Gap\left(\frac{1}{K}\sum_{k=1}^{K} \boldsymbol{z}_k\right) \leq \sup_{\boldsymbol{u} \in \mathcal{Z}} \frac{1}{K}\sum_{k=1}^{K} \langle G(\boldsymbol{z}_k), \boldsymbol{z}_k - \boldsymbol{u} \rangle.$$

*Proof.* For all $\boldsymbol{z} \in \mathcal{Z}$, $f(\boldsymbol{z}^{\boldsymbol{x}}, \boldsymbol{u}^{\boldsymbol{y}})$ is concave in $\boldsymbol{u}^{\boldsymbol{y}}$ and $-f(\boldsymbol{u}^{\boldsymbol{x}}, \boldsymbol{z}^{\boldsymbol{y}})$ is concave in $\boldsymbol{u}^{\boldsymbol{x}}$. Therefore, gap$(\boldsymbol{z}, \boldsymbol{u})$ is concave in $\boldsymbol{u}$ for every $\boldsymbol{z}$ and we have

$$\begin{aligned}
\text{gap}(\boldsymbol{z}, \boldsymbol{u}) &\leq \text{gap}(\boldsymbol{z}, \boldsymbol{z}) + \langle \nabla_{\boldsymbol{u}} \text{gap}(\boldsymbol{z}, \boldsymbol{z}), \boldsymbol{u} - \boldsymbol{z} \rangle \\
&= \langle \nabla_{\boldsymbol{u}} \text{gap}(\boldsymbol{z}, \boldsymbol{z}), \boldsymbol{u} - \boldsymbol{z} \rangle \\
&= \langle -G(\boldsymbol{z}), \boldsymbol{u} - \boldsymbol{z} \rangle \\
&= \langle G(\boldsymbol{z}), \boldsymbol{z} - \boldsymbol{u} \rangle.
\end{aligned}$$

Similarly, gap$(\boldsymbol{z}; \boldsymbol{u})$ is convex in $\boldsymbol{z}$ for every $\boldsymbol{u}$. Therefore,

$$\text{gap}\left(\frac{1}{K}\sum_{k=1}^{K} \boldsymbol{z}_k; \boldsymbol{u}\right) \leq \frac{1}{K}\sum_{k=1}^{K} \text{gap}(\boldsymbol{z}_k; \boldsymbol{u}) \leq \frac{1}{K}\sum_{k=1}^{K} \langle G(\boldsymbol{z}_k), \boldsymbol{z}_k - \boldsymbol{u} \rangle,$$

where the first inequality follows from convexity in $\boldsymbol{z}$ and the second inequality from above's result. Taking the supremum over the inequality yields the result. $\square$

The two divergences that we will use in the stochastic mirror descent algorithm for the $\ell_\infty$-$\ell_1$ convex-concave min-max problem are the following:

1. given the euclidean distance $h(\boldsymbol{x}) = \frac{1}{2}\|\boldsymbol{x}\|_2^2$, we obtain the divergence $V_{\boldsymbol{x}}(\boldsymbol{x}') = \frac{1}{2}\|\boldsymbol{x} - \boldsymbol{x}'\|_2^2$;

2. given $h(\boldsymbol{y}) = \sum_i y_i \log y_i$, we obtain the Kullback-Leibler divergence $V_{\boldsymbol{y}}(\boldsymbol{y}') = \sum y_i \log\left(\frac{y_i'}{y_i}\right)$.

**Theorem 3.** *Given a $\ell_\infty$-$\ell_1$ convex-concave-min-max problem 3, desired accuracy $\epsilon$, $(v^{\boldsymbol{x}}, \|\cdot\|_2)$-bounded estimators $\tilde{g}^{\boldsymbol{x}}$ of $g^{\boldsymbol{x}}$, and $(\frac{2v^{\boldsymbol{y}}}{\epsilon}, v^{\boldsymbol{y}}, \|\cdot\|_{\Delta^m}^2)$-bounded estimators $\tilde{g}^{\boldsymbol{y}}$ of $g^{\boldsymbol{y}}$. Algorithm 1 with choice of parameters $\eta_{\boldsymbol{x}} \leq \frac{\epsilon}{4v^{\boldsymbol{x}}}$, $\eta_{\boldsymbol{y}} \leq \frac{\epsilon}{4v^{\boldsymbol{y}}}$ outputs an $\epsilon$-approximate optimal solution within any iteration number $T \geq \max\{\frac{16nb^2}{\epsilon\eta_x}, \frac{8\log(m)}{\epsilon\eta_y}\}$.*

*Proof.* Note that $V_{\boldsymbol{x}}$ is 1-strongly convex. Since $\eta_{\boldsymbol{y}} \leq \frac{\epsilon}{4v^{\boldsymbol{y}}}$, we have that

$$\|\eta^{\boldsymbol{y}} \tilde{g}_t^{\boldsymbol{y}}\|_\infty \leq \frac{\epsilon}{4v^{\boldsymbol{y}}} \cdot \|\tilde{g}_t^{\boldsymbol{y}}\|_\infty \leq \frac{\epsilon}{4v^{\boldsymbol{y}}} \cdot \frac{2v^{\boldsymbol{y}}}{\epsilon} = \frac{1}{2}.$$

Hence, by Lemma 1 and Lemma 2 in Jin & Sidford (2020) we know that

$$\sum_{t \in [T]} \langle \eta^{\boldsymbol{x}} \tilde{g}_t^{\boldsymbol{x}}, \boldsymbol{x}_t - \boldsymbol{x} \rangle \leq V_{\boldsymbol{x}_1}(\boldsymbol{x}) + \frac{\eta^{\boldsymbol{x}2}}{2} \sum_{t \in [T]} \|\tilde{g}_t^{\boldsymbol{x}}\|_2^2,$$

$$\sum_{t \in [T]} \langle \eta^{\boldsymbol{y}} \tilde{g}_t^{\boldsymbol{y}}, \boldsymbol{y}_t - \boldsymbol{y} \rangle \leq V_{\boldsymbol{y}_1}(\boldsymbol{y}) + \frac{\eta^{\boldsymbol{y}2}}{2} \sum_{t \in [T]} \|\tilde{g}_t^{\boldsymbol{y}}\|_{\boldsymbol{y}_t}^2.$$

Now, define $\hat{g}_t^{\boldsymbol{x}} := g_t^{\boldsymbol{x}} - \tilde{g}_t^{\boldsymbol{x}}$, $\hat{g}_t^{\boldsymbol{y}} := g_t^{\boldsymbol{y}} - \tilde{g}_t^{\boldsymbol{y}}$, and the sequences $\hat{\boldsymbol{x}}_1, ..., \hat{\boldsymbol{x}}_T$ and $\hat{\boldsymbol{y}}_1, ..., \hat{\boldsymbol{y}}_T$ by

$$\hat{\boldsymbol{x}}_1 = \boldsymbol{x}_1, \ \hat{\boldsymbol{x}}_{t+1} = \operatorname*{arg\,min}_{\boldsymbol{x} \in \mathbb{B}_b^n} \langle \eta^{\boldsymbol{x}} \hat{g}_t^{\boldsymbol{x}}, \boldsymbol{x} \rangle + V_{\hat{\boldsymbol{x}}_t}(\boldsymbol{x}),$$

$$\hat{\boldsymbol{y}}_1 = \boldsymbol{y}_1, \ \hat{\boldsymbol{y}}_{t+1} = \operatorname*{arg\,min}_{\boldsymbol{y} \in \Delta^m} \langle \eta^{\boldsymbol{y}} \hat{g}_t^{\boldsymbol{y}}, \boldsymbol{y} \rangle + V_{\hat{\boldsymbol{y}}_t}(\boldsymbol{y}).$$

In a similar way to $\eta^{\boldsymbol{y}} g_t^{\boldsymbol{y}}$, we can bound the $\ell_\infty$-norm of $\eta^{\boldsymbol{y}} \hat{g}_t^{\boldsymbol{y}}$

$$\|\eta^{\boldsymbol{y}} \hat{g}_t^{\boldsymbol{y}}\|_\infty \leq \|\eta^{\boldsymbol{y}} \tilde{g}_t^{\boldsymbol{y}}\|_\infty + \|\eta^{\boldsymbol{y}} g_t^{\boldsymbol{y}}\|_\infty = \|\eta^{\boldsymbol{y}} \tilde{g}_t^{\boldsymbol{y}}\|_\infty + \|\mathbb{E}[\eta^{\boldsymbol{y}} \tilde{g}_t^{\boldsymbol{y}}]\|_\infty \leq 2\|\eta^{\boldsymbol{y}} \tilde{g}_t^{\boldsymbol{y}}\|_\infty \leq 1.$$

Therefore, using the lemmas as above we get

$$\sum_{t \in [T]} \langle \eta^{\boldsymbol{x}} \hat{g}_t^{\boldsymbol{x}}, \boldsymbol{x}_t - \boldsymbol{x} \rangle \leq V_{\boldsymbol{x}_1}(\boldsymbol{x}) + \frac{\eta^{\boldsymbol{x}2}}{2} \sum_{t \in [T]} \|\hat{g}_t^{\boldsymbol{x}}\|_2^2,$$

$$\sum_{t \in [T]} \langle \eta^{\boldsymbol{y}} \hat{g}_t^{\boldsymbol{y}}, \boldsymbol{y}_t - \boldsymbol{y} \rangle \leq V_{\boldsymbol{y}_1}(\boldsymbol{y}) + \frac{\eta^{\boldsymbol{y}2}}{2} \sum_{t \in [T]} \|\hat{g}_t^{\boldsymbol{y}}\|_{\boldsymbol{y}_t}^2.$$

Since $g_t^{\boldsymbol{x}} = \hat{g}_t^{\boldsymbol{x}} + \tilde{g}_t^{\boldsymbol{x}}$ and $g_t^{\boldsymbol{y}} = \hat{g}_t^{\boldsymbol{y}} + \tilde{g}_t^{\boldsymbol{y}}$,

$$\sum_{t \in [T]} [\langle g_t^{\boldsymbol{x}}, \boldsymbol{x}_t - \boldsymbol{x} \rangle + \langle g_t^{\boldsymbol{y}}, \boldsymbol{y}_t - \boldsymbol{y} \rangle]$$

$$= \sum_{t \in [T]} \left[ \frac{1}{\eta^{\boldsymbol{x}}} \langle \eta^{\boldsymbol{x}} \tilde{g}_t^{\boldsymbol{x}}, \boldsymbol{x}_t - \boldsymbol{x} \rangle + \frac{1}{\eta^{\boldsymbol{y}}} \langle \eta^{\boldsymbol{y}} \tilde{g}_t^{\boldsymbol{y}}, \boldsymbol{y}_t - \boldsymbol{y} \rangle \right] + \sum_{t \in [T]} \left[ \frac{1}{\eta^{\boldsymbol{x}}} \langle \eta^{\boldsymbol{x}} \hat{g}_t^{\boldsymbol{x}}, \hat{\boldsymbol{x}}_t - \boldsymbol{x} \rangle + \frac{1}{\eta^{\boldsymbol{y}}} \langle \eta^{\boldsymbol{y}} \hat{g}_t^{\boldsymbol{y}}, \hat{\boldsymbol{y}}_t - \boldsymbol{y} \rangle \right]$$

$$+ \sum_{t \in [T]} [\langle \hat{g}_t^{\boldsymbol{x}}, \boldsymbol{x}_t - \hat{\boldsymbol{x}}_t \rangle + \langle \hat{g}_t^{\boldsymbol{y}}, \boldsymbol{y}_t - \hat{\boldsymbol{y}}_t \rangle]$$

$$= \frac{1}{\eta^{\boldsymbol{x}}} \sum_{t \in [T]} [\langle \eta^{\boldsymbol{x}} \tilde{g}_t^{\boldsymbol{x}}, \boldsymbol{x}_t - \boldsymbol{x} \rangle] + \frac{1}{\eta^{\boldsymbol{x}}} \sum_{t \in [T]} [\langle \eta^{\boldsymbol{x}} \hat{g}_t^{\boldsymbol{x}}, \hat{\boldsymbol{x}}_t - \boldsymbol{x} \rangle] + \frac{1}{\eta^{\boldsymbol{y}}} \sum_{t \in [T]} [\langle \eta^{\boldsymbol{y}} \tilde{g}_t^{\boldsymbol{y}}, \boldsymbol{y}_t - \boldsymbol{y} \rangle] + \frac{1}{\eta^{\boldsymbol{y}}} \sum_{t \in [T]} [\langle \eta^{\boldsymbol{y}} \hat{g}_t^{\boldsymbol{y}}, \hat{\boldsymbol{y}}_t - \boldsymbol{y} \rangle]$$

$$+ \sum_{t \in [T]} [\langle \hat{g}_t^{\boldsymbol{x}}, \boldsymbol{x}_t - \hat{\boldsymbol{x}}_t \rangle + \langle \hat{g}_t^{\boldsymbol{y}}, \boldsymbol{y}_t - \hat{\boldsymbol{y}}_t \rangle]$$

$$\leq \frac{2}{\eta^{\boldsymbol{x}}} V_{\boldsymbol{x}_1}(\boldsymbol{x}) + \frac{\eta^{\boldsymbol{x}}}{2} \sum_{t \in [T]} [\|\tilde{g}_t^{\boldsymbol{x}}\|_2^2 + \|\hat{g}_t^{\boldsymbol{x}}\|_2^2] + \sum_{t \in [T]} \langle \hat{g}_t^{\boldsymbol{x}}, \boldsymbol{x}_t - \hat{\boldsymbol{x}}_t \rangle$$

$$+ \frac{2}{\eta^{\boldsymbol{y}}} V_{\boldsymbol{y}_1}(\boldsymbol{y}) + \frac{\eta^{\boldsymbol{y}}}{2} \sum_{t \in [T]} [\|\tilde{g}_t^{\boldsymbol{y}}\|_{\boldsymbol{y}_t}^2 + \|\hat{g}_t^{\boldsymbol{y}}\|_{\boldsymbol{y}_t}^2] + \sum_{t \in [T]} \langle \hat{g}_t^{\boldsymbol{y}}, \boldsymbol{y}_t - \hat{\boldsymbol{y}}_t \rangle.$$

Consider the operator $G(\boldsymbol{z}) := [g^{\boldsymbol{x}}(\boldsymbol{x}, \boldsymbol{y}), -g^{\boldsymbol{y}}(\boldsymbol{x}, \boldsymbol{y})]$. As min-max problem 3 is convex in its first argument and concave in its second argument, by Lemma 3, if we show that

$$\sup_{\boldsymbol{u} \in \mathcal{Z}} \frac{1}{T} \sum_{t \in [T]} \langle G(\boldsymbol{z}_t), \boldsymbol{z}_t - \boldsymbol{u} \rangle \leq \epsilon,$$

we obtain that $\mathrm{Gap}(\frac{1}{T}\sum_{t=1}^{T}\boldsymbol{z}_t) \leq \epsilon$. Now take the expectation on both sides:

$$\mathbb{E}\sup_{\boldsymbol{u}\in\mathcal{Z}}\frac{1}{T}\sum_{t\in[T]}\langle G(\boldsymbol{z}_t),\boldsymbol{z}_t-\boldsymbol{u}\rangle = \mathbb{E}\frac{1}{T}\sup_{(\boldsymbol{x},\boldsymbol{y})}\left[\sum_{t\in[T]}\langle g^{\boldsymbol{x}}(\boldsymbol{x},\boldsymbol{y}),\boldsymbol{x}_t-\boldsymbol{x}\rangle + \sum_{t\in[T]}\langle g^{\boldsymbol{y}}(\boldsymbol{x},\boldsymbol{y}),\boldsymbol{y}_t-\boldsymbol{y}\rangle\right]$$

$$\overset{(i)}{\leq}\frac{1}{T}\mathbb{E}\sup_{(\boldsymbol{x},\boldsymbol{y})}\left[\frac{2}{\eta^{\boldsymbol{x}}}V_{\boldsymbol{x}_1}(\boldsymbol{x}) + \frac{\eta^{\boldsymbol{x}}}{2}\sum_{t\in[T]}[\|\tilde{g}_t^{\boldsymbol{x}}\|_2^2 + \|\hat{g}_t^{\boldsymbol{x}}\|_2^2]\right.$$

$$\left. + \frac{2}{\eta^{\boldsymbol{y}}}V_{\boldsymbol{y}_1}(\boldsymbol{y}) + \frac{\eta^{\boldsymbol{y}}}{2}\sum_{t\in[T]}[\|\tilde{g}_t^{\boldsymbol{y}}\|_2^2 + \|\hat{g}_t^{\boldsymbol{y}}\|_2^2]\right]$$

$$\overset{(ii)}{\leq}\frac{1}{T}\mathbb{E}\sup_{(\boldsymbol{x},\boldsymbol{y})}\left[\frac{2}{\eta^{\boldsymbol{x}}}V_{\boldsymbol{x}_1}(\boldsymbol{x}) + \eta^{\boldsymbol{x}}\sum_{t\in[T]}\|\tilde{g}_t^{\boldsymbol{x}}\|_2^2 + \frac{2}{\eta^{\boldsymbol{y}}}V_{\boldsymbol{y}_1}(\boldsymbol{y}) + \eta^{\boldsymbol{y}}\sum_{t\in[T]}\|\tilde{g}_t^{\boldsymbol{y}}\|_2^2\right]$$

$$\overset{(iii)}{\leq}\sup_{\boldsymbol{x}}\frac{2}{\eta^{\boldsymbol{x}}T}V_{\boldsymbol{x}_1}(\boldsymbol{x}) + \eta^{\boldsymbol{x}}v^{\boldsymbol{x}} + \sup_{\boldsymbol{y}}\frac{2}{\eta^{\boldsymbol{y}}T}V_{\boldsymbol{y}_1}(\boldsymbol{y}) + \eta^{\boldsymbol{y}}v^{\boldsymbol{y}}$$

$$\overset{(iv)}{\leq}\frac{4nb^2}{\eta^{\boldsymbol{x}}T} + \eta^{\boldsymbol{x}}v^{\boldsymbol{x}} + \frac{2\log m}{\eta^{\boldsymbol{y}}T} + \eta^{\boldsymbol{y}}v^{\boldsymbol{y}}$$

$$\overset{(v)}{\leq}\epsilon,$$

where in $(i)$ we used that $\mathbb{E}[\langle\hat{g}_t^{\boldsymbol{x}},\boldsymbol{x}_t-\hat{\boldsymbol{x}}_t\rangle \mid 1,...,T] = \mathbb{E}[\langle\hat{g}_t^{\boldsymbol{y}},\boldsymbol{y}_t-\hat{\boldsymbol{y}}_t\rangle \mid 1,...,T] = 0$; $(ii)$ $\mathbb{E}[\|\hat{g}_t^{\boldsymbol{x}}\|_2^2] \leq \mathbb{E}[\|\tilde{g}_t^{\boldsymbol{x}}\|_2^2]$ and $\mathbb{E}[\sum_i[\hat{y}_t]_i[\hat{g}_t^{\boldsymbol{y}}]_i^2] \leq \mathbb{E}[\sum_i[\hat{y}_t]_i[\tilde{g}_t^{\boldsymbol{y}}]_i^2]$ due to $\mathbb{E}[(X-\mathbb{E}[X])^2] \leq \mathbb{E}[X^2]$; $(iii)$ due to the assumptions on the estimators; $(iv)$ by properties of KL-divergence and that $\frac{1}{2}\|\boldsymbol{x}-\boldsymbol{x}_0\|_2^2 \leq 2nb^2$; $(v)$ the choice of $\eta^{\boldsymbol{x}} = \frac{\epsilon}{4v^{\boldsymbol{x}}}$, $\eta^{\boldsymbol{y}} = \frac{\epsilon}{4v^{\boldsymbol{y}}}$, and $T \geq \max\{\frac{16nb^2}{\epsilon\eta^{\boldsymbol{x}}}, \frac{8\log m}{\epsilon\eta^{\boldsymbol{y}}}\}$.    $\square$

**Theorem 2.** *Given $\epsilon \in (0,1)$, Algorithm 1 with step-size*

$$\eta^{(\boldsymbol{c},\boldsymbol{u})} = \frac{\epsilon}{4v^{(\boldsymbol{c},\boldsymbol{u})}}, \quad \eta^{\boldsymbol{\mu}} = \frac{\epsilon}{4v^{\boldsymbol{\mu}}},$$

*and gradient estimators defined in equation 1 and 2 finds an expected $\epsilon$-approximate solution within any iteration number*

$$T \geq \max\left\{\mathcal{O}\left(\frac{\alpha^2|\mathcal{S}|^3|\mathcal{A}|^2}{\epsilon^2}\right), \mathcal{O}\left(\frac{|\mathcal{S}||\mathcal{A}|\log(|\mathcal{S}||\mathcal{A}|)}{\epsilon^2}\right)\right\}.$$

*Proof.* This follows directly from the bounds of the gradient estimators and Theorem 3.    $\square$

## B    $\epsilon$-approximate solutions

**Proposition 4.** *Let $((\boldsymbol{c}^\epsilon,\boldsymbol{u}^\epsilon)\boldsymbol{\mu}^\epsilon)$ be an $\epsilon$-approximate solution for $RLfD_\alpha$, where $\boldsymbol{\mu}^\epsilon$ induces a policy $\pi_{\boldsymbol{\mu}^\epsilon} \in \Pi_0$ defined by $\pi_{\boldsymbol{\mu}^\epsilon}(a|s) = \frac{\boldsymbol{\mu}(a,s)}{\sum_{a'}\boldsymbol{\mu}(s,a)}$. It then holds that*

$$\mathbb{E}\left[\alpha\|\boldsymbol{c}^\epsilon-\hat{\boldsymbol{c}}\|_2^2 + \rho_{\boldsymbol{c}^\epsilon}(\pi_E) - \rho_{\boldsymbol{c}^\epsilon}(\pi_A)\right] \leq \epsilon + \alpha\|\boldsymbol{c}_A-\hat{\boldsymbol{c}}\|_2^2 + \rho_{\boldsymbol{c}_A}(\pi_E) - \rho_{\boldsymbol{c}_A}^*,$$

*where $((\boldsymbol{c}_A,\boldsymbol{u}_A),\boldsymbol{\mu}_A)$ denotes the optimal solution for $RLfD_\alpha$ and $\pi_A$ is the policy induced by $\boldsymbol{\mu}_A$.*

*Proof.* As $((\boldsymbol{c}^\epsilon,\boldsymbol{u}^\epsilon)\boldsymbol{\mu}^\epsilon)$ is an $\epsilon$-approximate solution we know that

$$\mathbb{E}\left[\mathcal{L}((\boldsymbol{c}^\epsilon,\boldsymbol{u}^\epsilon),\boldsymbol{\mu}_A) - \mathcal{L}((\boldsymbol{c}_A,V_{\boldsymbol{c}_A}^{\pi_{\boldsymbol{\mu}^\epsilon}}),\boldsymbol{\mu}^\epsilon)\right] \leq \epsilon.$$

Moreover,

$$\mathcal{L}((\boldsymbol{c}^\epsilon, \boldsymbol{u}^\epsilon), \boldsymbol{\mu}_A) - \mathcal{L}((\boldsymbol{c}_A, \boldsymbol{V}_{\boldsymbol{c}_A}^{\pi_{\boldsymbol{\mu}^\epsilon}}), \boldsymbol{\mu}^\epsilon)$$

$$= \alpha\|\boldsymbol{c}^\epsilon - \hat{\boldsymbol{c}}\|_2^2 - \alpha\|\boldsymbol{c}_A - \hat{\boldsymbol{c}}\|_2^2 + \langle \boldsymbol{\mu}_{\pi_E} - \boldsymbol{\mu}_A, \boldsymbol{c}^\epsilon - \boldsymbol{T}_\gamma^\top \boldsymbol{u}^\epsilon \rangle - \langle \boldsymbol{\mu}_{\pi_E} - \boldsymbol{\mu}^\epsilon, \boldsymbol{c}_A - \boldsymbol{T}_\gamma^\top \boldsymbol{V}_{\boldsymbol{c}_A}^{\pi_{\boldsymbol{\mu}^\epsilon}} \rangle$$

$$\stackrel{(i)}{=} \alpha\|\boldsymbol{c}^\epsilon - \hat{\boldsymbol{c}}\|_2^2 - \alpha\|\boldsymbol{c}_A - \hat{\boldsymbol{c}}\|_2^2 + \rho_{\boldsymbol{c}^\epsilon}(\pi_E) - \rho_{\boldsymbol{c}^\epsilon}(\pi_A) - \rho_{\boldsymbol{c}_A}(\pi_E) + \rho_{\boldsymbol{c}_A}(\pi_{\boldsymbol{\mu}^\epsilon})$$

$$\stackrel{(ii)}{\geq} \alpha\|\boldsymbol{c}^\epsilon - \hat{\boldsymbol{c}}\|_2^2 - \alpha\|\boldsymbol{c}_A - \hat{\boldsymbol{c}}\|_2^2 + \rho_{\boldsymbol{c}^\epsilon}(\pi_E) - \rho_{\boldsymbol{c}^\epsilon}(\pi_A) - \rho_{\boldsymbol{c}_A}(\pi_E) + \rho_{\boldsymbol{c}_A}(\pi_A),$$

where $(i)$ follows from $\langle \boldsymbol{\mu}_1 - \boldsymbol{\mu}_2, -\boldsymbol{T}_\gamma^\top \boldsymbol{u} \rangle = \langle \boldsymbol{\nu}_0, \boldsymbol{u} \rangle - \langle \boldsymbol{\nu}_0, \boldsymbol{u} \rangle = 0$ and Lemma 2 (Appendix B.1) in Kamoutsi et al. (2021) and $(ii)$ from $\rho_{\boldsymbol{c}_A}^* = \rho_{\boldsymbol{c}_A}(\pi_A) \leq \rho_{\boldsymbol{c}_A}(\pi)$ for any policy $\pi \in \Pi_0$. Rearranging these terms we arrive to

$$\mathbb{E}\left[\alpha\|\boldsymbol{c}^\epsilon - \hat{\boldsymbol{c}}\|_2^2 + \rho_{\boldsymbol{c}^\epsilon}(\pi_E) - \rho_{\boldsymbol{c}^\epsilon}(\pi_A)\right] \leq \epsilon + \alpha\|\boldsymbol{c}_A - \hat{\boldsymbol{c}}\|_2^2 + \rho_{\boldsymbol{c}_A}(\pi_E) - \rho_{\boldsymbol{c}_A}^*$$

$\square$

