# OpenReview forum: "Inverse Reinforcement Learning via Inverse Optimization"
_TMLR — Rejected by TMLR_

### Review · Reviewer_REeY · 2025-03-24

**Summary Of Contributions:**

The authors draw connections between Inverse reinforcement learning and inverse optimisation. Especially they claim to extend the work of Kamoutsi et al. (2021) by optimising a regularised objective instead of that of the original work. In the experiments they show how $\alpha=0$ (Kamoutsi et al.) fails to recover the correct cost function, but the introduced regularised version succeeds.

**Audience:**

Yes

**Claims And Evidence:**

No

**Requested Changes:**

Mathematical concepts that are wrong or unclear:
- Normalized value function is first a vector, one line further it is a function?
- $(\text{RL}_{\text{c}})$: Maybe $s_0 \sim \nu_0$ ?
- After $RL_c$:, you say that this is equivalent to $min_\pi <v_0, V_c^{\pi}>$, so an inner product between a distribution and a (value) function? Or a vector and a vector? Probably the latter but you should be careful on the definitions and the words. Same with $<\mu_\pi, c>$ later.
- The discussion on $T_\gamma \mu$ and $\mathcal{F}$ has no interpretation and would be beneficial for the reader to introduce some interpretation of these concepts. Is $\frac{1}{1-\gamma}(B - \gamma P)\mu$ missing a transpose?
- In MDP-P_c I don't understand why $T_\gamma \mu = \nu_0$ makes sense. I think it relates to the previous point where no intuition is given of $T_\gamma$.
- FOP: What is $\theta$?
- FOP: What is $\Gamma$?
- FOP: What is $X(\theta)$?
- Overall the notation and the story in the background follow strictly two papers from Kamoutsi et al. and Chan et al. with little overlap in notation, I think you should present the method in your own words, or explicitly state that "We strictly use the same notation as in Kamoutsi et al. as we are going to draw strong connections to their work", and then stick with it. Presented as is, the manuscript feels like a copy-paste from these works.
- IRL-IO: Why select such $u$ that $F(c)$ is minimised? What is $F(c)$?
- Eq. (2) and algorithm: What is $e^{(s,a)}$?
- Experiments, first paragraph: cost or reward, cannot be both or same thing?
- Experiments: Figure 5 is referred, others are not, probably a mistake in \ref
- Experiments: I find it difficult to understand what are the expert demonstrations in these experiments? Or does "access to a generative-model oracle of the expert’s occupancy
measure" really mean that we need access to expert policy instead of demonstrations from it? Please clarify.
- Interpretation of the results should be further discussed, what are the implications of the results?
- Experiments: What is the difference between $c_{\text{true}}$ and $\hat{c}$?
- Experiments: Discussion why does regularization help the convergence? (measured in terms of duality gap in Fig. 5)

**Strengths And Weaknesses:**

Strengths:
The paper does empirical evaluation in Gridworld environment, which is something that not all theoretical work do.

Weaknesses:
I think the biggest weakness of the manuscript is the presentation. The manuscript attempts for mathematical rigour but many of the equations have no explanations, no motivation, and use notation that is not explained or not commonly known in the field. There is a strong dependency to the cited papers, to the extent that the concepts (at least in the background) are basically copied from either Kamoutsi et al. or Chan et al. with little added insight to these works. I am going to list some of the problems in the "Requested Changes" part of my review, but am sceptical if they will have a substantial effect on my final decision as the cluttered notation makes it almost impossible to understand the rest of the paper.

If some of the theorems and their proofs have novelty that the authors want to claim as contribution, they should be properly referred and discussed in the main paper. It should also be discussed how they complement the earlier works.

---

> ### Author Response · Authors · 2025-04-10
> **Response**
>
> $\textbf{Answers to requested changes:}$
>
> We thank the reviewer for their insightful comments. In response, we have clarified our problem setting and added Proposition 3, Proposition 4, as well as Figure 1 and Figure 2 to better explain and motivate the formulations of $IO-IRL_{\hat{\mathbf{c}}}$ and $\text{IO-AL}_{\alpha}$. Additionally, we have improved the clarity of the notation used throughout the manuscript. Below, we detail each of these changes:
>
> RC1-RC3: we clarified that the value function is represented as a vector and made the use of inner products more precise throughout the manuscript.
>
> RC4-5: We provided an intuitive explanation for Proposition 1 that explains $\mathcal{F}$ and its interpretation as the set of occupancy measures. Moreover, $\frac{1}{1 - \gamma}(B - \gamma P)\mu$ is not missing a transpose; however, Kamoutsi et al. denote it with a transpose, but transpose it again later for some calculations. Furthermore, we included the interpretation of $T_{\gamma} \mu = \nu_0$ as the Markov property. Moreover, we explained that the constraints of $\text{MDP-P}_{\mathbf{c}}$ enforce $\mu$ to be within the set of occupancy measures.
>
> RC6-9 and RC11: We specified the notation that was missing in the Notation subsection and the inverse optimization notation was explained more clearly in section 2.4.
>
> RC10: The IRL-IO problem is the general inverse optimization problem for finding the cost function of an MDP. As we want to find a $c$ instead of an $u$, the way of choosing cost functions in the inverse-feasible set should just depend on $c$ and the way we do it is with a generic $F(c)$ that has not yet been instantiated. To see the instantiated problem observe $\text{IRL-IO}_{\hat{\mathbf{c}}}$.
>
> RC12-13: We consistently used the term cost throughout the manuscript, removing references to reward for clarity and coherence. Additionally, we corrected the href formatting error.
>
> RC14: Generative-model oracle means that we can query the expert's occupancy measure and it will return us a state-action pair $(s,a) \sim \boldsymbol{\mu}_{\pi_E}$. Hence, we do not have access to demonstrations, but an oracle of the expert's policy.
>
> RC15: In our experiments, $\hat{\mathbf{c}}$ denotes the estimated cost vector, while $\mathbf{c}_{\text{true}}$ represents the true cost function that the expert is assumed to optimize. Specifically, the true cost function assigns a value of 1 to obstacle cells (red), -1 to the goal cell (green), and 0 to all other cells (white). In contrast, the estimated cost function $\hat{\mathbf{c}}$ assigns 0 to all cells by default, with 50\% of the obstacles set to 1 and 50\% of the goal cells set to -1.
>
> RC16: We added comments discussing the impact of regularization on convergence. However, we would like to clarify that the duality gap figure is intended to illustrate the empirical behavior of regularization on convergence, rather than to theoretically characterize it. As established in Theorem 2, regularization negatively affects convergence in theory and our experimental results show that stronger regularization leads to a slower reduction in the duality gap during the initial iterations of the algorithm.

---

> > ### Comment · Reviewer_REeY · 2025-04-14
> >
> > Thank you for the response. I think it clarified my understanding, and the updates improved the paper.

---

### Review · Reviewer_pupg · 2025-03-25

**Summary Of Contributions:**

This paper proposes a new algorithm for IRL. The authors address the fact that there exists a feasible set of cost functions compatible with the (optimal or suboptimal) expert's policy by making a specific choice of cost function from such set, depending on two parameters $\widehat{c}$ and $\alpha$. Then, the authors solve this min-max optimization problem via stochastic mirror descent in a finite-sample regime, and, in this way, claim to be able to compute $\epsilon$-correct estimates of the specific cost function formulated earlier. Finally, some numerical simulations are conducted to validate the results.

**Audience:**

No

**Broader Impact Concerns:**

None.

**Claims And Evidence:**

No

**Requested Changes:**

I believe that this paper requires substantial changes to clarify the weaknesses that I mentioned above. Specifically, I do believe that the learning target considered in the paper (i.e., the cost function in (IO-AL$\alpha$)), does not represent a meaningful contribution for the IRL and AL communities because it is not motivated why that cost function should be interesting. Given this issue, and the problematic of Theorem 2, I think that this paper should be rejected for this venue.

**Strengths And Weaknesses:**

STRENGTHS:

The paper studies the IRL and AL problems that are interesting and of practical importance for the community.

WEAKNESSES:
- The authors propose to address the well-known IRL identifiability problem by selecting the cost function that results from the optimization problem in (IO-AL$\alpha$). However, it is unclear why this cost function should be better than the others, and, also, what relationship there is between this cost function and the expert's policy. These issues are crucial, and, in my opinion, represent strong limits to the contributions of the paper.
- Theorem 2 provides convergence guarantees for Algorithm 1, that works in a finite sample regime, i.e., as mentioned by the authors, in a setting where the transition model $T$ and expert's distribution $\mu_E$ are *unavailable*. However, it does not provide a PAC guarantee, i.e., it does not hold with high probability, but it holds with probability 1, which makes me doubt a lot on the correctness of the result or the problem setting considered.

MINOR WEAKNESSES:
- It is unclear how the parameters $\widehat{c}$ and $\alpha$ should be selected.
- Why do you require that $v_0(s)>0$ strictly for every possible state? This is kind a strong assumption.
- Why do you require that $\gamma > 0$ strictly?
- In Section 2.1 you use the Kronecker delta $\delta_{s,t}$ without defining. I think you should define it explicitly.
- In section 2.2 you talk about "reward function", but you should say "cost function".
- For Proposition 2 there is no reference nor proof.
- Theorem 1 should be made clearer that is comes from Kamoutsi et al. (2021).
- at beginning of section 3.2, I do not understand why you can search for $u$ in the box. What is the meaning of $u$?

---

> ### Author Response · Authors · 2025-04-10
> **Response**
>
> $\textbf{Answers to weaknesses: }$
> We thank the reviewer for the comments. These comments allowed us to clarify both the notation and the presentation of our results, particularly Theorem 2.
>
> W1: Regarding, the relationship between the cost function found and the expert's policy, we included Figure 1 and Figure 2 that provide intuition for the solutions to $IRL-IO_{\hat{\mathbf{c}}}$ and $IO-AL_{\alpha}$. Furthermore, we added Proposition 3 that characterizes the optimal solution to $\text{IO-AL}_{\alpha}$ and Proposition 4 that explains that an $\epsilon$-approximate solution, i.e. the output of the algorithm proposed, is at most $\epsilon$ worse than the optimal solution in expectation.
>
> W2: Regarding Theorem 2, we have clarified that the output of the algorithm $((\mathbf{c}^\epsilon, \mathbf{u}^\epsilon), \boldsymbol{\mu}^\epsilon) $ is an $\epsilon$-approximate solution, which means that it satisfies $$E\left[\text{Gap}((\mathbf{c}^\epsilon, \mathbf{u}^\epsilon), \boldsymbol{\mu}^\epsilon)\right] \leq \epsilon,$$
> i.e. we provide a bound on the expected value of the solution.
>
> $\textbf{Answers to minor weaknesses: }$
>
> MW1: The parameter $\hat{c}$ should be chosen with respect to the prior beliefs of the learner regarding the cost function and it does not need to be accurate. Regarding $\alpha$, it should be nonnegative and is an hyperparameter that depends on each problem instance.
>
> MW2: We require that $\nu_0(s) > 0$ for Proposition 1 to make sense. It ensures that the induced policy is well defined.
>
> MW3: We requiere that $\gamma > 0$ strictly because if $\gamma$ where to be $0$, then all policies would have total expected cost $0$.
>
> MW4-5: We applied the proposed changes.
>
> MW6: We included a reference for Proposition 2.
>
> MW7: We clarified this.
>
> MW8: We added the interpretation for $\mathbf{u}$ in the preliminaries. It is the decision variable for $\text{MDP-D}_{\mathbf{c}}$ and corresponds to the value function. Lemma 3 in the Appendix explains why we can search for $\mathbf{u}$ in the box.

---

### Review · Reviewer_Gk2U · 2025-03-27

**Summary Of Contributions:**

The authors study inverse reinforcement learning in infinite-horizon tabular MDPs with assumptions on sample-access to the expert policy transitions (such that they can continuously query the expert during training) and a generative model. In this setting, they make two contributions:

First, they show that when the expert policy is optimal in the underlying MDP, the feasible set of the Learning from Demonstrations (LfD) formalism of Kamoutsi et al (2021) is equivalent to the inverse-feasible set of the classic linear programming formulation of MDPs. Therefore, they relate inverse RL problem (particularly the LfD problem) to inverse optimization on the original LP framework.

Their second contribution is an efficient stochastic saddle-point optimization scheme (based on Jin and Sidford 2020) to solve a variant of the unconstrained inverse optimization (IO) problem: learn the true cost and value function for which the expert behaviour is optimal. Here, they modify the original IO problem to account for when the expert behaviour is sub-optimal and it is possible to estimate the true cost vector. They provide an upperbound on the duality gap of their method on the resulting saddle-point problem, and validate this result with experiments on a small grid-world problem.

**Audience:**

Yes

**Claims And Evidence:**

Yes

**Requested Changes:**

Please see below some questions, as well as typos and comments:

**Questions**

1. What is the intuition behind equation $IO-AL_{\alpha}$? Can the authors clarify their intuition behind reweighting the original objective with the temperature parameter $(1-\alpha)$, rather than simply “adding regularization” to the original objective. Differently from the latter approach, the former introduces a trade-off between learning the true cost that induces expert behavior and fitting the estimated cost.
2. From my understanding, the primary objective of inverse RL is to learn the true cost function, given an expert policy. Therefore, being able to obtain an accurate estimate $\hat{c}$ of $c_{true}$ would make the IRL problem non-existent. With this in mind, is it even possible to learn $c_{true}$ with a bad cost estimate and sample-access to a suboptimal expert policy? More importantly, from a theoretical perspective, can the bound on the duality gap in Theorem 1 be translated to small suboptimality for the apprentice?

**Typos and comments**
1. Section 1.1
    * Sentence 2: “...probability simplex over elements n elements” should be “... probability simplex over n elements ”.
    * Sentence 2: The notation in the simplex should match the number “n” of elements.
2. Section 2.1
    * Paragraph 6 (just before prop. 1): The use of $t$ to denote a state is confusing. I would suggest using $s’$ instead. Also, can the authors clarify that $\delta_{s,s’}$ is the indicator or entirely replace it with the indicator.
    * Proposition 1:
        * The induced policy $\pi_{\mu}$ is not well-defined with the current definition, as the denominator could be zero for some state.
3. Section 3
    * Paragraph 1: Should $RL_c$ be $RL_{c_{true}}$ and $MDP-P_c$ be $MDP-P_{c_{true}}$?
4. Section 3.1
    * Paragraph 2: I believe the correct phrase should be “... represents the apprentice state-action visitation probability”, not “...represents the apprentice policy”
5. Section 5.2
    * All figure hyperlinks incorrectly point to Figure 5.
    * For the grid-world environment, how are the rewards fixed for cells without obstacles and non-terminal cells?
    * For clarity, can the authors kindly highlight how the apprentice policy in section 5.2 is extracted from the solution of Algorithm 1?

**Strengths And Weaknesses:**

**Strengths**

The paper is mostly well-written and to the best of my knowledge properly cites related literature. I also find the idea of connecting inverse RL to inverse optimization interesting.
\
\
**Weaknesses**

I have some doubts regarding the novelty and completeness of contributions made in the paper.
Regarding the first contribution: it appears that inverse optimization and inverse RL have been considered in prior works, particularly Erkin et al. (2010). This paper specifically aims to solve an inverse RL problem via inverse optimization – thereby already establishing a connection between both areas. More so, the authors first result in Theorem 1 is relatively trivial in my opinion – as the conclusion can be easily derived from earlier equations.

Regarding the second contribution: The main result in theorem 2 is not exactly conclusive. Theorem 2 only includes a bound on the expected duality gap. This result does not say anything about the quality of the learned cost $c^{\varepsilon}$ – which would be an anticipated outcome for the inverse RL problem, or the suboptimality of the learned policy – which would be an expected outcome in the LfD or apprenticeship learning setting. I kindly invite the authors to check out Lemma 10 of Jin and Sidford (2020) or Proposition 5 of Kamoutsi et al (2021) in order to arrive at a truly complete contribution.

In addition to the above remarks, it is unclear how the work compares with related literature on the specific setting of inverse RL in tabular MDPs with sample-access to transitions from a sub-optimal expert policy and a generative model.

---

> ### Author Response · Authors · 2025-04-10
> **Response**
>
> $\textbf{Answers to questions: }$
> We thank the reviewer for their valuable questions and insights. These comments helped us to clarify the problem formulation and provide new results.
>
> Q1: This question played a pivotal role in helping us reframe our approach. In particular, it led us to introduce the regularization term without the $(1-\alpha)$ term, which in turn enabled the development of Proposition 4. Furthermore, we included figures 1 and 2 that explain the intuition regarding solutions to problems $IO-IRL_{\hat{\mathbf{c}}}$ and $\text{IO-AL}_{\alpha}$.
>
> Q2: We clarified that the cost vector estimate $\hat{\mathbf{c}}$ does not need to be accurate. For the $IO-IRL_{\hat{\mathbf{c}}}$ problem, we project this estimate to the inverse-feasible set. Hence, we choose an optimal cost vector that is closest to $\hat{\mathbf{c}}$. On the other hand, the solution to $IO-AL_{\alpha}$ balances the difference between the learned cost vector $c_A$ and the estimate $\hat{\mathbf{c}}$ and the difference between the expert policy $\pi_E$ and the apprentice policy $\pi_A$. Regarding, learning a true cost vector with a bad cost estimate and sample-access to a suboptimal expert policy, Proposition 3 characterizes optimal solutions to  $IO-AL_{\alpha}$ and Proposition 4 provides bounds for an expected $\epsilon$-approximate solution where the suboptimality of the expert is measured with respect to the optimal solution. Note that in the case of a suboptimal expert we do not aim to find the true cost vector because the expert is not optimal for any cost vector.
>
> $\textbf{Answers to typos and comments: }$
> We thank the reviewer for identifying the typos, we have corrected them in the revised manuscript. Regarding the comment on Proposition 1, in the introduction we assume $\nu_0 > 0$, hence the constraints in $\text{MDP-P}_{\mathbf{c}}$ ensure that the denominator is strictly positive. Furthermore, we specified the costs fixed for cells without obstacles and non-terminal cells and highlighted how we extracted the apprentice policy at the beginning of the experimental section.

---

### Decision · Action_Editor_wk7E · 2025-05-13

**Recommendation:** Reject

**Comment:**

The reviewers appreciated the paper attempt to connect iIRL with optimization, recognizing its potential on the theory side. However, they expressed several concerns, especially about clarity, realism of the assumptions, and experimental support. Based on these concerns, my recommendation is to reject the paper at this time. However, embedding the reviewers' suggestions may significantly improve the paper for future submissions.

**Audience:**

At least some individuals in TMLR’s audience would likely be interested in the findings of this paper. However, broader interest might be limited due to the preliminary nature of empirical results and due to the strong assumptions.

**Claims And Evidence:**

The paper provides evidence for some its claims, but, according to the reviewers they are not fully convincing due to the following limitations:

- Theoretical evidence: Theorem 2 provides a bound to the duality gap but this does not directly guarantee the quality of the resulting policy, thus, limiting the practical implications of this result.

- Assumptions: some important assumptions, like access to valid occupancy measures and prior over the cost function, seem to be quite strong and not sufficiently justified.

- Empirical evidence: the experimental campaign is limited to a simple grid-world environment and lacks comparisons to relevant baselines. Thus, it provides only partial support to the method effectiveness.

**Resubmission Of Major Revision:**

The authors may consider submitting a major revision at a later time.